# Towards Robust Gradient Regularization with Central-difference and Momentum Lookahead

## Abstract

Sharpness-Aware Minimization (SAM), which can be extended to a form of gradient regularization, is an effective technique for improving generalization by guiding optimizers towards flat minima through parameter perturbations. However, extending such regularization strategies to multi-step settings often leads to instability, where naive iterative updates degrade rather than enhance generalization. To overcome this limitation, we propose Central-difference Momentum Lookahead Regularization (CMLR), a framework that performs momentum lookahead through central-difference probing of the loss landscape. By constructing the perturbation direction from symmetric gradient evaluations, CMLR realizes a momentum lookahead update that is inherently more robust and exhibits reduced variance, while requiring no additional gradient evaluations. This design ensures smooth optimization trajectories and reliable improvements at low computational cost. We conduct a comprehensive theoretical analysis of CMLR and its foundational versions (CLR, LR), presenting spectral analysis results, variance reduction analysis, and establishing formal convergence guarantees, particularly under a momentum strategy. Empirically, we demonstrate that CMLR consistently improves generalization across diverse architectures and datasets.

## 1 Introduction

A fundamental objective in deep learning is to discover model parameters that achieve strong generalization beyond mere minimization of training loss. This pursuit has motivated a line of work culminating in the Sharpness-Aware Minimization (SAM) algorithm (Foret et al., 2020), which explicitly seeks parameters in flat regions of the loss landscape. SAM has demonstrated strong generalization across numerous tasks (Chen et al., 2021; Zhang et al., 2021), sparking follow-up studies to improve its behavior (Bartlett et al., 2023; Du et al., 2022; Jiang et al., 2023; Li & Giannakis, 2024; Sun et al., 2023; Wen et al., 2023).

However, SAM comes with a puzzling catch: trying to solve its inner optimization problem more accurately, particularly with multi-step methods, often makes the final model generalize worse, not better (Foret et al., 2020; Andriushchenko & Flammarion, 2022; Kim et al., 2023b; Mordido et al., 2024). A new perspective, viewing SAM through the lens of Gradient Regularization (GR), helps explain why (Barrett & Dherin, 2021; Smith et al., 2021; Zhao et al., 2022; Reizinger & Huszár, 2023). This view reveals that SAM is essentially using a simple forward-difference approximation of the Hessian (a strategy we call FR) (Zhao et al., 2022; Karakida et al., 2023). This type of approximation is known to be unstable, which likely causes the gradient estimates to become noisy and unreliable during the multi-step ascent process, ultimately hurting performance (Liu et al., 2022b). We will therefore classify this generalization-enhancing regularization as a form of gradient regularization and proceed to analyze its different variants.

Since the instability of SAM originates from its rough finite difference approximation, from the perspective of feature decomposition, this can be explained as the gradient oscillating near the saddle point (Kim et al., 2023a; Tan et al., 2024b), leading to a tendency towards model suboptimal. Therefore, a more accurate central difference scheme and forward-looking mechanism can ensure that the parameter update process can more firmly escape the saddle point, thereby

guiding the model to improve generalization. We first replace the forward-difference with a more precise central-difference scheme (CR). To test this hypothesis, we measure the stability of the optimization path by calculating the cosine similarity between consecutive update directions (clipped to $[0, 1]$). Our experiments reveal a clear result: a higher concentration of similarity scores away from 0 strongly correlates with better generalization, and our CR method produces a much smoother optimization trajectory with consistently higher similarity, as shown in Figure 1.

A stable update is a great starting point, but we also want the benefits of multi-step optimization that have proven successful in other SAM variants (Mordido et al., 2024; Tan et al., 2024a; Yu et al., 2024). To achieve this, we embed our stable CR update within a momentum lookahead mechanism (ML). However, this combination would be far too slow for practical use. The key to making it efficient is our final contribution: a lightweight momentum lookahead mechanism, which allows us to approximate future gradient information with no extra computational cost.

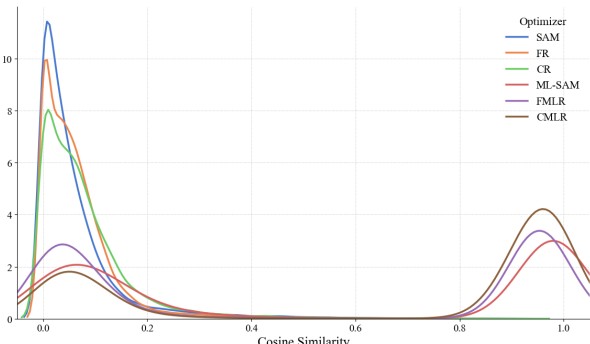

Figure 1: Distribution of optimizer update direction stability (smoothed for visual clarity). The figure illustrates the distribution of cosine similarity between consecutive update directions $d_k$ during training for different optimizers using ResNet-18 trained on CIFAR-10. For LA-SAM, FMLR, and CMLR, the parameter K was set to 2.

The power of this complete approach (result in Central-difference Momentum Lookahead gradient Regularization, CMLR; Forward-difference Momentum Lookahead gradient Regularization, FMLR; Momentum Lookahead SAM, ML-SAM) is validated by our final results in Figure 1. The test accuracies show a clear hierarchy (CMLR > FMLR > CR > FR > SAM > ML-SAM). Our results highlight a key finding: while momentum lookahead helps the general FR framework, it actually hurts performance in the specific case of SAM (ML-SAM), consistent with (Kim et al., 2023b; Mordido et al., 2024). This suggests the paradox is at its worst when the update, by setting the regularization strength exactly equal to the size of the parameter perturbation neighborhood, completely ignores the stabilizing influence of the base gradient. This is why CMLR succeeds; because it starts with a much more stable central-difference foundation, it can fully take advantage of the momentum lookahead mechanism.

Building on this foundation, we propose a multi-step generalization of this framework and introduce a lightweight inner-loop momentum lookahead strategy. To avoid excessive computational overhead, we incorporate a momentum lookahead mechanism that approximates future perturbations using previously computed gradients, thus requiring no additional gradient evaluations. The resulting method, Central-difference Momentum Lookahead Regularization (CMLR), promotes the training stability of central-difference GR while enabling efficient multi-step updates.

The main contributions of this paper are summarized as follows:

- We propose CMLR, which is an optimizer that uses central differencing to perform momentum smoothing more stably in the loop within the Lookhead, avoiding the problem of generalization degradation caused by multi-step exploration.
- We provide a comprehensive theoretical analysis for our proposed gradient regularization and momentum lookahead strategies.
- We demonstrate through extensive experiments on various model architectures and diverse datasets that CMLR significantly improves generalization performance.

## 2 BACKGROUND AND RELATED WORK

To understand our new perspective, let's first consider the standard approach. Typically, training a neural network means finding the weights $w \in \mathcal{W} \subset \mathbb{R}^d$ that minimize the empirical risk on a training dataset $\mathcal{S} = \{(x_i, y_i)\}_{i=1}^n$ with $(x_i, y_i)$ drawn i.i.d. from an underlying distribution $\mathcal{D}$ over $\mathcal{X} \times \mathcal{Y}$. Using the per-data-point loss function $l : \mathcal{W} \times \mathcal{X} \times \mathcal{Y} \to \mathbb{R}$, this goal is expressed as: $\min_w \mathcal{L}(w) = \frac{1}{n} \sum_{i=1}^n l(w, x_i, y_i)$. We assume that the loss function $\mathcal{L}$ is twice differentiable

throughout the paper. For notational simplicity, unless otherwise specified, we use the shorthand $\nabla \mathcal{L}(w)$ for $\nabla_w \mathcal{L}(w)$ and $\| \cdot \|$ for the L2 norm, $\| \cdot \|_2$.

## 2.1 Flat Minima

The geometric properties of the loss landscape are widely considered to be intrinsically linked to a model's generalization ability. A central tenet, originating from early theoretical work and bolstered by extensive empirical studies (Hochreiter & Schmidhuber, 1997; Keskar et al., 2016), posits that models converging to flat, wide minima generalize better than those in sharp, narrow valleys. While this principle has been challenged by theoretical counter-examples showing that sharpness can be sensitive to reparameterization (Dinh et al., 2017), a strong correlation between measures of flatness and generalization performance is consistently observed in practice (Jiang et al., 2019). Among these, Sharpness-Aware Minimization (SAM) (Foret et al., 2020) has emerged as a state-of-the-art approach, whose core idea is to connect the training loss $\mathcal{L}(w)$ with the model's generalization bound. Instead of minimizing the loss at a single point $w$, SAM seeks parameters that reside in a neighborhood of uniformly low loss by solving a min-max problem:

$$\min_w \mathcal{L}_{\text{SAM}}(w) \quad \text{where} \quad \mathcal{L}_{\text{SAM}}(w) \triangleq \max_{\|\varepsilon\|_p \leq \rho} \mathcal{L}(w + \varepsilon). \tag{1}$$

As solving the inner maximization exactly is intractable, SAM approximates the solution with a first-order perturbation $\hat{\varepsilon}$, In the standard setting where the L2 norm is used ($p = 2$), this perturbation aligns with the direction of the gradient $\nabla \mathcal{L}(w)$. The final update is then performed using the gradient at this perturbed point $w + \hat{\varepsilon}$:

$$\nabla \mathcal{L}_{SAM} = \nabla \mathcal{L}(w + \hat{\varepsilon}) \quad \text{where} \quad \hat{\varepsilon} = \rho \frac{\nabla \mathcal{L}(w)}{\|\nabla \mathcal{L}(w)\|}. \tag{2}$$

## 2.2 The Non-robustness Paradox of SAM

While SAM excels at finding flat minima, it presents a key paradox: trying to solve its inner optimization more accurately with multi-step methods often hurts generalization instead of helping it (Foret et al., 2020; Andriushchenko & Flammarion, 2022; Mordido et al., 2024). The consensus points to gradient instability as the main cause; during these multi-step updates, the model's parameters stray too far from their starting point (Kim et al., 2023b; Mordido et al., 2024).

This fragility is well-documented. Recent studies have shown that SAM is sensitive to noisy, high-variance gradients (Hassan et al., 2025) and operates on a dynamic "edge of robustness" that depends on the gradient norm (Long & Bartlett, 2024). The theory of Gradient Regularization (GR) offers a deeper explanation. It reveals that SAM's core mathematical step—a forward-difference approximation—is inherently unstable, struggling with both very small and very large perturbation sizes (Karakida et al., 2023). The multi-step approach effectively forces SAM into this problematic small-step scenario, which likely explains the performance drop.

## 2.3 Robustness and Efficiency Enhancements

Research on improving SAM has largely focused on tackling two major challenges:

The first challenge, non-robustness, arises because noisy gradient estimates can make the training process erratic. A popular strategy to counteract this is to "smooth out" the trajectory. Many methods take inspiration from the Lookahead optimizer (Zhang et al., 2019), which works by averaging recent model weights to prevent drastic jumps. Following this principle, approaches like SALA (Tan et al., 2024a) and Lookbehind-SAM (Mordido et al., 2024) have shown that using lookahead ideas or aggregating information from past steps leads to a much more stable training process. Other methods attack the problem more directly, either by adding explicit curvature regularization to the objective (Wu et al., 2024) or by cleaning up the gradient statistics themselves (Hassan et al., 2025).

The second challenge is efficiency. Standard SAM is expensive, essentially doubling the workload by requiring two gradient computations for every single update. This has spurred the development of a family of "efficient SAM" variants. Their common strategy is to cleverly reuse or approximate gradients to cut down the computational overhead, making the benefits of sharpness-aware training more accessible (Du et al., 2021; 2022; Mi et al., 2022; Liu et al., 2022a; Jiang et al., 2023; Wang et al., 2024; Becker et al., 2024).

## 3 METHODOLOGY

Our approach begins with a general framework for sharpness-aware methods that is based on Gradient Regularization. Into this framework, we incorporate a lookahead mechanism to help steer the optimization process. To keep things efficient, we also add an momentum accumulation strategy that prevents high computational costs. Together, these components form our algorithm: Central-difference Momentum Lookahead Regularization (CMLR).

### 3.1 GRADIENT REGULARIZATION (GR)

Gradient Regularization (GR) is a general approach to optimization where the training objective is modified to penalize the gradient norm, thereby encouraging convergence to flatter regions of the loss landscape (Barrett & Dherin, 2021; Smith et al., 2021; Zhao et al., 2022; Karakida et al., 2023; Reizinger & Huszár, 2023):

$$\min_w \mathcal{L}_{\text{GR}}(w) \triangleq \mathcal{L}(w) + \lambda \|\nabla \mathcal{L}(w)\|. \tag{3}$$

Calculating the exact gradient for the regularization term in Equation 3 requires expensive second-order information, specifically a Hessian-vector product (Hochreiter & Schmidhuber, 1997; Jastrzebski et al., 2021):

$$\nabla \|\nabla \mathcal{L}(w)\| = \frac{\nabla^2 \mathcal{L}(w) \nabla \mathcal{L}(w)}{\|\nabla \mathcal{L}(w)\|}. \tag{4}$$

While this term can be computed exactly using Double Backpropagation (DB) in modern frameworks like PyTorch, it's known to be less computationally efficient than using a finite-difference approximation (Karakida et al., 2023). Recent analyses have formally established that the SAM update is equivalent to a first-order, forward-difference approximation of this term (Zhao et al., 2022; Karakida et al., 2023).

To formalize this connection, let's consider the exact gradient of the GR objective from Equation 3. Let $v = \frac{\nabla \mathcal{L}(w)}{\|\nabla \mathcal{L}(w)\|}$ be the normalized gradient direction. The core idea of the forward-difference approach is to approximate this costly second-order term using only first-order information. This is achieved using the following approximation, derived from the first-order Taylor expansion of the gradient function:

$$\nabla^2 \mathcal{L}(w)v = \frac{\nabla \mathcal{L}(w + \rho v) - \nabla \mathcal{L}(w)}{\rho} + O(\rho). \tag{5}$$

By substituting this approximation into the exact GR gradient, we obtain the update rule for Forward-Difference Gradient Regularization (FR):

$$\nabla \mathcal{L}_{\text{FR}}(w) \triangleq \nabla \mathcal{L}(w) + \lambda \left( \frac{\nabla \mathcal{L}(w + \rho v) - \nabla \mathcal{L}(w)}{\rho} \right)$$

$$= \left( 1 - \frac{\lambda}{\rho} \right) \nabla \mathcal{L}(w) + \frac{\lambda}{\rho} \nabla \mathcal{L}(w + \rho v). \tag{6}$$

This formulation expresses the update as a weighted average of the gradients at the original point $w$ and the perturbed point $w + \rho v$. Notably, in the standard setting where the regularization strength is set to be equal to the perturbation radius: $\lambda = \rho$, the FR update simplifies to exactly $\nabla \mathcal{L}(w + \rho v)$. This is precisely the gradient update rule used by SAM, thus confirming its role as a specific instance of the GR framework.

The forward-difference scheme provides a first-order approximation with $\mathcal{O}(\rho)$ error. To achieve a more accurate estimation of the regularized gradient, we instead employ the second-order central-difference approximation. Its $\mathcal{O}(\rho^2)$ accuracy is formally derived from the Taylor series expansion of $\nabla \mathcal{L}(w \pm \rho v)$:

$$\nabla^2 \mathcal{L}(w)v = \frac{\nabla \mathcal{L}(w + \rho v) - \nabla \mathcal{L}(w - \rho v)}{2\rho} + O(\rho^2). \tag{7}$$

Substituting this into the GR gradient objective yields our proposed Central-difference Gradient Regularization (CR) update rule:

$$\nabla \mathcal{L}_{\text{CR}}(w) \triangleq \nabla \mathcal{L}(w) + \frac{\lambda}{2\rho} \left( \nabla \mathcal{L}(w + \rho v) - \nabla \mathcal{L}(w - \rho v) \right). \tag{8}$$

The core idea of this formulation is to shift the update's focus. It relies more on the standard gradient at the original point $w$ with a central-difference scheme. This stands in sharp contrast to adversarial methods like FR (Zhao et al., 2022) and CR-SAM (Wu et al., 2024), which still primarily rely on the perturbed gradient.

---

**Algorithm 1** CMLR

---

**Require:** Loss function $\mathcal{L}$, training data $G$, base optimizer $\mathcal{A}$, inner steps $K$, outer steps $T$, initial slow weights $w_0$, batch size $b$, inner step size $\eta_{t,k}$, outer step size $\alpha$, perturbation radius $\rho$, smoothing factor $\beta_s$ and $\beta_e$.
1: **for** $t = 0, 1, 2, \ldots, T-1$ **do**
2:      Synchronize weights: $w_{t,0} \leftarrow w_t$.
3:      Sample a mini-batch $B \subset G$ with size $b$.
4:      Initialize perturbation: $v_0, g_0 \leftarrow \nabla\mathcal{L}(w_{t,0})$ on $B$.
5:      **for** $k = 0, 1, 2, \ldots, K-1$ **do**
6:          Normalize perturbation: $\hat{v}_k \leftarrow \frac{v_k}{\|v_k\|}$.
7:          Compute central-difference gradients in parallel:
8:          Parallel do:
9:          $g_k^+ \leftarrow \nabla\mathcal{L}(w_{t,k} + \rho\hat{v}_k)$ on $B$.
10:         $g_k^- \leftarrow \nabla\mathcal{L}(w_{t,k} - \rho\hat{v}_k)$ on $B$.
11:         End parallel.
12:         Update direction: $d_k \leftarrow \frac{\rho+\lambda}{2\rho}g_k^+ + \frac{\rho-\lambda}{2\rho}g_k^-$.
13:         Update fast weights: $w_{t,k+1} \leftarrow \mathcal{A}(w_{t,k}, \eta_{t,k}, d_k)$.
14:         Anticipating the next perturbation vector:
15:         $\beta_k \leftarrow \beta_s + (\beta_e - \beta_s) \cdot \frac{k}{K-1}$.
16:         $v_{k+1} \leftarrow \beta_k\hat{v}_k + \frac{(1-\beta_k)}{2}\left(\frac{g_k^+}{\|g_k^+\|} + \frac{g_k^-}{\|g_k^-\|}\right)$.
17:      **end for**
18:      Update slow weights (Lookahead step): $w_{t+1} \leftarrow w_t + \alpha(w_{t,K} - w_t)$
19: **end for**
20: **return** $w_T$
**Ensure:** CMLR trained model.

---

### 3.2 MOMENTUM LOOKAHEAD MECHANISM

The primary challenge of our Central-difference (CR) update (Equation 8) is its high computational cost, requiring three gradient computations per step. To make the computation more feasible and the momentum update more robust, we introduce a stable momentum accumulation mechanism. This approach avoids explicitly computing the base gradient, relying instead solely on the ascent and descent perturbation gradients. We then place it within the two-timescale Lookahead framework (Zhang et al., 2019; Yu et al., 2024) to improve training stability. Prior works have suggested that Lookahead-based variants of SAM are particularly effective in boosting generalization (Tan et al., 2024a; Mordido et al., 2024), which further motivates our integration. Although Lookahead's inner loop performs multiple updates, the computation remains feasible since no extra overhead is introduced inside the inner iterations. This highlights that CMLR represents a principled and effective integration, rather than a simple stacking of separate components.

At each inner-loop step $k$, the core efficiency gain comes from bypassing the explicit computation of the base gradient $\nabla\mathcal{L}(w_k)$. We approximate it using the average of the two perturbed gradients, $g_k^+ \triangleq \nabla\mathcal{L}(w_k + \rho v_k)$ and $g_k^- \triangleq \nabla\mathcal{L}(w_k - \rho v_k)$, which are already required for the central difference. This leads to a final update direction $d_k$ that depends only on these two gradients, which can be computed in parallel:

$$
\begin{aligned}
d_k &= \frac{g_k^+ + g_k^-}{2} + \frac{\lambda}{2\rho}(g_k^+ - g_k^-) \\
&= \frac{\rho+\lambda}{2\rho}g_k^+ + \frac{\rho-\lambda}{2\rho}g_k^-.
\end{aligned}
\tag{9}
$$

To prepare for the subsequent inner-loop step $k+1$ without additional cost, we anticipate the next perturbation vector $v_{k+1}$ by using a moving average of the previously computed normalized gradi-

ents, a strategy inspired by recent work on efficient SAM variants (Wang et al., 2024),

$$v_{k+1} = \beta_k v_k + \frac{(1 - \beta_k)}{2} \left( \frac{g_k^+}{\|g_k^+\|} + \frac{g_k^-}{\|g_k^-\|} \right). \tag{10}$$

The fast weights are then updated using this direction, $w_{t,k+1} = w_{t,k} - \eta_{t,k} d_k$. After $K$ inner-loop steps, the slow weights are updated in the direction of the final fast weights, consistent with the Lookahead framework. The complete procedure, which synthesizes these components, is detailed in Algorithm 1, In addition, we design a process toy example of the algorithm in Figure 2.

The divergent outcomes of applying Lookahead to FR and SAM highlight a key mechanic. The FR update, $d_k = (1 - \frac{\lambda}{\rho})g_k + \frac{\lambda}{\rho}g_k^+$, retains the base gradient $g_k$ as a stable anchor when $\lambda \neq \rho$, which allows Lookahead to effectively smooth the update trajectory. In the specific case of SAM where $\lambda = \rho$ or $d_k = (1 - \frac{\lambda}{\rho})g_k^+ + \frac{\lambda}{\rho}g_k^+ = g_k^+ (k \geq 1)$ in FMLR, this anchor vanishes, leaving the update to depend solely on the noisy perturbed gradient $g_k^+$. Without this stabilizing reference, Lookahead fails to improve performance. Our CR, by contrast, provides a robust, symmetric foundation that consistently synergizes with the Lookahead framework.

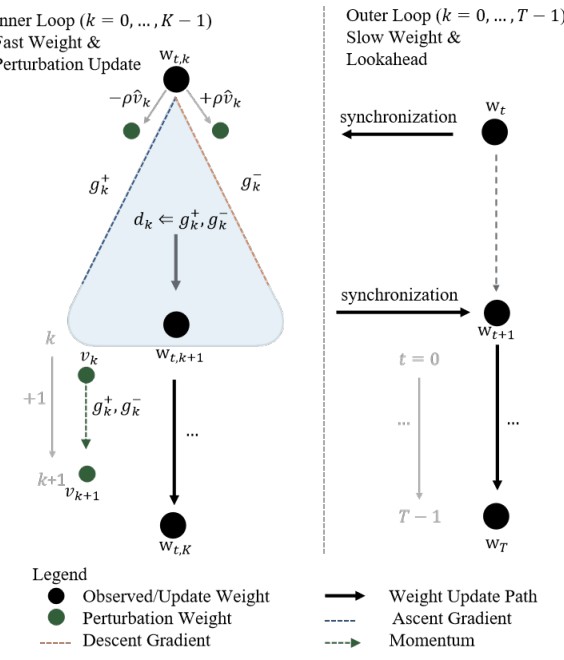

Figure 2: Toy example analysis of CMLR

This effectively collapses the method into the standard Momentum Lookahead-SAM (ML-SAM), where the regularizing influence of the base gradient is lost in subsequent inner-loop steps. Our CR approach, by contrast, maintains its two-term structure due to the symmetric nature of its probes, ensuring the regularization is applied consistently. This synergy makes the lookahead framework an ideal counterpart to our central-difference strategy.

## 4 CONVERGENCE ANALYSIS

In this section, we present the convergence analysis and variance spectral analysis conclusions of CMLR, and place the complete convergence analysis and variance analysis of the basic version algorithms (LR, CLR) in Appendix B and Appendix C. Specifically, we analyze variance reduction in the noisy quadratic model. To demonstrate the stabilizing effect of the Lookahead mechanism, we adopt the regularized objective $\mathcal{L}(w) + \frac{\lambda}{2}\|\nabla\mathcal{L}(w)\|^2$ a standard and analytically tractable formulation that simplifies the mathematics by using the squared gradient norm while preserving the core optimization goal (Barrett & Dherin, 2021; Smith et al., 2021; Karakida et al., 2023).

### 4.1 CONVERGENCE FOR GENERAL NON-CONVEX OBJECTIVES

To unify the dual-timescale loops into a single time frame, we employ a two-time-scale analysis (Borkar, 1997; Nedic & Ozdaglar, 2009; Wang et al., 2020). This results in a version of the formal algorithm that differs only slightly in its presentation. Specifically, to unify the inner and outer loops, the slow-weight update is synchronized with the final fast-weight update of the inner loop. This change, driven by the need to synchronize the inner and outer loop weights, leads to an apparent one-step reduction in the inner loop, but it does not substantially affect the final weight update. To facilitate understanding, we provide a simplified version of the pseudocode in Algorithm 2 (Appendix A). Finally, to formalize the synchronization scheme for the inner- and outer-loop weights, we re-index the iterates as follows: $y_s = w_{t,k}$ represents the inner-loop "fast" weights with a single global step counter $s = tK + k$, and $w_s = y_{\lfloor s/K \rfloor K}$ are the outer-loop "slow" weights. Here, $\lfloor \cdot \rfloor$ denotes the floor function, which rounds its argument down to the nearest integer. The "middle"

weights are defined as $\hat{y}_s = \alpha y_s + (1 - \alpha)w_s$ which performs the update $\hat{y}_{s+1} = \hat{y}_s - \alpha \eta_s d(y_s, \xi_s)$. Our analysis views CLR as a stochastic gradient method on an implicit, non-smooth objective function: $F(w) = \mathcal{L}(w) + \lambda \|\nabla \mathcal{L}(w)\|$.

We make the following standard assumptions for our analysis. We first provide a brief analysis of the basic version (CLR), and then present the convergence analysis of the CMLR algorithm with momentum mechanism with its full proof detailed in Appendix B and C.

**Theorem 4.1** (Convergence of CMLR for Non-Convex Objectives). *Under the same conditions in Theorem B.6, the iterative sequence generated by CMLR satisfies:*

$$\liminf_{s \to \infty} \mathbb{E}[\|\nabla F(\hat{y}_s)\|^2] = \mathcal{O}(\lambda). \tag{11}$$

It is worth noting that the momentum strategy does not fundamentally alter the convergence guarantee under gradient regularization, which remains of the order $O(\lambda)$. Instead, its primary impact lies in refining the higher-order constant terms. For instance, two similar conclusions suggest that momentum strategies are more likely to provide more accurate approximations for these high-order constants, which are related to $\lambda, \rho, L_1$ and $L_2$. This is exemplified by the replacement of $C_1$ with $\tilde{C}_1$ in our analysis in Appendix C.

### 4.2 Noisy Quadratic Analysis with Gradient Regularization

To understand the convergence properties of our proposed method, we extend the noisy quadratic analysis framework from Schaul et al. (2013); Wu et al. (2018); Zhang et al. (2019). We analyze the standard noisy quadratic model, $\hat{\mathcal{L}}(x) = \frac{1}{2}(x - c)^\top H(x - c)$, with $c \sim \mathcal{N}(x^*, \Sigma)$, where $H, \Sigma \in \mathbb{R}^{d \times d}$ are assumed to be diagonal and $x, c \in \mathbb{R}^d, x^* = 0$.

Here, we obtained the variance reduction conclusion of CMLR using spectral analysis, which is detailed in Appendix C. The $v_{\text{LR}}$ mentioned in the theorem comes from a simple extension of a previous work, as shown in Appendix B.

**Theorem 4.2** (Variance Reduction with CMLR). *Fix an eigenpair $(q, \mu)$ of $H$, with scalar projections $x_{t,k} = q^\top w_{t,k}$, $c_{t,k} = q^\top c$, and $\hat{v}_{k,q} = q^\top \hat{v}_k$. Let $a \triangleq 1 - \eta\mu$, define $A_{\text{eff}} \triangleq (1 - \alpha) + \alpha a^K, B_{\text{eff}} \triangleq \alpha \eta \mu$. Under Assumption B.2, Assumption B.4, $\|\nabla \mathcal{L}\| \geq g_{\min}$ and condition in Lemma E.1, the steady-state variance of CMLR satisfies*

$$v_{\text{CMLR}} \leq v_{\text{LR}}(1 + \sqrt{\tau})^2 + \mathcal{O}\left(\frac{\rho^4 \|H\|^4 + M}{g_{\min}^4}\right) \tag{12}$$

*where*

$$v_{\text{LR}} = \frac{\alpha^2(\eta\mu)^2\sigma^2 \dfrac{1 - a^{2K}}{1 - a^2}}{1 - A_{\text{eff}}^2}, \qquad \tau = 2d\frac{\lambda^2(1 - \beta)}{g_{\min}^2(1 + \beta)}, \tag{13}$$

Compared with the initial gradient-regularization estimator, the Lookahead step in LR already acts as a variance-reduction mechanism, so $v_{\text{LR}}$ is smaller than the variance of the initial gradient-regularization scheme. In Theorem 4.2, the additional factor $(1 + \sqrt{\tau})^2$ in the bound for $v_{\text{CMLR}}$ arises from the momentum prediction strategy and corresponds to only a controllable $\mathcal{O}(\lambda)$ variance inflation, since $(1 + \sqrt{\tau})^2 = 1 + \mathcal{O}(\lambda)$. Combining these observations, by choosing $\lambda$ sufficiently small, the steady-state variance of CMLR in our bound can be made smaller than that of the initial gradient-regularization estimator. It should be noted that, while Lookahead itself contributes to variance reduction through averaging, the variance reduction effect emphasized in our analysis is mainly attributed to the central-difference gradient regularization, rather than the Lookahead gradient regularization alone.

## 5 Experiments

To demonstrate the broad applicability of CMLR, we evaluate SGD, AdamW, SAM, CR-SAM (Wu et al., 2024), Lookbehind-SAM (Mordido et al., 2024), GSAM (Wang et al., 2024), FMLR and CMLR on the CIFAR-10 and CIFAR-100 datasets using the following models which include widely-used CNNs such as ResNet-18 (He et al., 2016), VGG-16 (Simonyan & Zisserman, 2014), WideResNet-28-10 (Zagoruyko & Komodakis, 2016), and PyramidNet-110 (Han et al., 2017), as well as popular Vision Transformers (ViT-Ti and ViT-S) (Dosovitskiy et al., 2020). To verify the efficiency of the algorithm, provide analysis results and time comparisons for the same number of

Table 1: Performance comparison of CMLR against baseline optimizers in CNN models (Test Accuracy %).

| Optimizer | CIFAR-10 (Test Accuracy %) | | | |
| --- | --- | --- | --- | --- |
| | ResNet-18 | WRN-28-10 | VGG-16-BN | PyramidNet-110 |
| SGD | $96.13_{\pm0.11}$ | $97.03_{\pm0.16}$ | $95.42_{\pm0.17}$ | $96.92_{\pm0.28}$ |
| SAM | $96.59_{\pm0.12}$ | $97.51_{\pm0.16}$ | $95.75_{\pm0.13}$ | $97.59_{\pm0.29}$ |
| CR-SAM | $96.79_{\pm0.14}$ | $97.71_{\pm0.12}$ | $95.95_{\pm0.16}$ | $97.79_{\pm0.21}$ |
| Lookbehind-SAM | $97.09_{\pm0.13}$ | $98.01_{\pm0.11}$ | $96.25_{\pm0.14}$ | $98.09_{\pm0.22}$ |
| GSAM | $97.34_{\pm0.12}$ | $98.26_{\pm0.13}$ | $96.51_{\pm0.14}$ | $98.35_{\pm0.23}$ |
| FMLR | $97.29_{\pm0.11}$ | $98.21_{\pm0.12}$ | $96.46_{\pm0.15}$ | $98.37_{\pm0.19}$ |
| **CMLR (Ours)** | $\mathbf{97.84}_{\pm0.11}$ | $\mathbf{98.63}_{\pm0.13}$ | $\mathbf{97.12}_{\pm0.18}$ | $\mathbf{98.91}_{\pm0.11}$ |

| Optimizer | CIFAR-100 (Test Accuracy %) | | | |
| --- | --- | --- | --- | --- |
| | ResNet-18 | WRN-28-10 | VGG-16-BN | PyramidNet-110 |
| SGD | $78.34_{\pm0.21}$ | $82.07_{\pm0.17}$ | $75.13_{\pm0.23}$ | $83.55_{\pm0.24}$ |
| SAM | $80.24_{\pm0.19}$ | $83.55_{\pm0.14}$ | $76.52_{\pm0.12}$ | $84.76_{\pm0.13}$ |
| CR-SAM | $80.42_{\pm0.18}$ | $83.70_{\pm0.13}$ | $76.72_{\pm0.14}$ | $84.99_{\pm0.14}$ |
| Lookbehind-SAM | $80.74_{\pm0.17}$ | $84.03_{\pm0.12}$ | $77.02_{\pm0.13}$ | $85.28_{\pm0.15}$ |
| GSAM | $80.86_{\pm0.16}$ | $84.35_{\pm0.13}$ | $77.33_{\pm0.15}$ | $85.56_{\pm0.13}$ |
| FMLR | $80.90_{\pm0.18}$ | $84.23_{\pm0.12}$ | $77.16_{\pm0.15}$ | $85.44_{\pm0.14}$ |
| **CMLR (Ours)** | $\mathbf{81.64}_{\pm0.14}$ | $\mathbf{84.84}_{\pm0.11}$ | $\mathbf{77.65}_{\pm0.13}$ | $\mathbf{86.07}_{\pm0.12}$ |

backpropagation iterations. To further validate the robustness and scalability of our method, we also extended our experiments to the Tiny-ImageNet (Le & Yang, 2015) datasets—where Tiny-ImageNet provides a 200-class, 64×64 downscaled subset of ImageNet for efficient benchmarking. Additionally, we evaluated algorithms on eight NLP tasks from the GLUE benchmark (Wang et al., 2018): CoLA, SST-2, MRPC, STS-B, QQP, MNLI, QNLI, and RTE, using a standard Transformer-based architecture (DistilBERT). Finally, we conducted ablation studies to analyze the algorithm's sensitivity to key hyperparameters, including regularization strength, the momentum accumulation Lookahead strategy, and the step size of the slow weights. A detailed report of these studies is provided in Appendix F.

## 5.1 CONVOLUTIONAL NEURAL NETWORK

For data augmentation, we first pad each training image by four pixels, take a random 32×32 crop, and apply a random horizontal flip. We then apply Cutout, masking a random 16×16 region of the image with zeros following DeVries & Taylor (2017).

Our experimental setup is configured as follows. First, under the standard setting for SAM and FR (Foret et al., 2020; Zhao et al., 2022; Li & Giannakis, 2023), we establish optimal general hyper-parameters, including the initial learning rate 0.05, weight decay 0.001, and perturbation magnitude ($\rho \in \{0.01, 0.05, 0.1\}$). The learning rate is updated following a cosine annealing schedule. Second, for the forward-difference gradient regularization optimizer, we adopt a grid search to determine the optimal value. For our proposed central-difference gradient regularization optimizer (CMLR), we configure the hyperparameters as follows: We hypothesize that this annealing strategy decreases variance in the later stages of training, thereby enhancing generalization. The hyperparameter $\alpha$ is selected from $[0.7, 1.0]$ and $\lambda$ is selected from $[0.05, 0.15]$. $\beta_k$, is smoothly annealed from 0.9 up to 0.99 over the course of training. For the integrated Lookahead mechanism, we provide the best results: K=10 for GSAM, FMLR and CMLR and K=5 for Lookbehind-SAM, and more detailed experimental results (K=2,5,10) are attached in the appendix. These relatively large values of $K$ are intentionally chosen to study whether Lookahead-style multi-step schemes can continue to improve generalization as $K$ increases, rather than to minimize wall-clock cost; a complementary comparison under matched gradient-evaluation budgets (equal compute) is reported later in our experiments. With the exception of the PyramidNet-110 model, which was trained for 300 epochs with a batch size of 256, all other models were trained for 200 epochs with a batch size of 128. The results are summarized in Table 1.

To validate our momentum lookahead strategy, we first conduct a fair comparison in terms of computational cost among several multi-step algorithms, namely Lookbehind-SAM, GSAM, FMLR and

CMLR. We evaluate the accuracy trends of different algorithms after an equal number of backprop-agation steps. Specifically, we plot the performance on ResNet-34/CIFAR-10 at 400, 800, 1200, 1600, and 2000 backpropagations. During the training middle process, it can be seen from Figure 3 that our proposed CMLR is almost always optimal and consistently achieves superior accuracy for the same computational budget. To further demonstrate this, we provide additional results on the ResNet-50/CIFAR-100 dataset in the appendix, which exhibit a similar trend (see Figure 7).

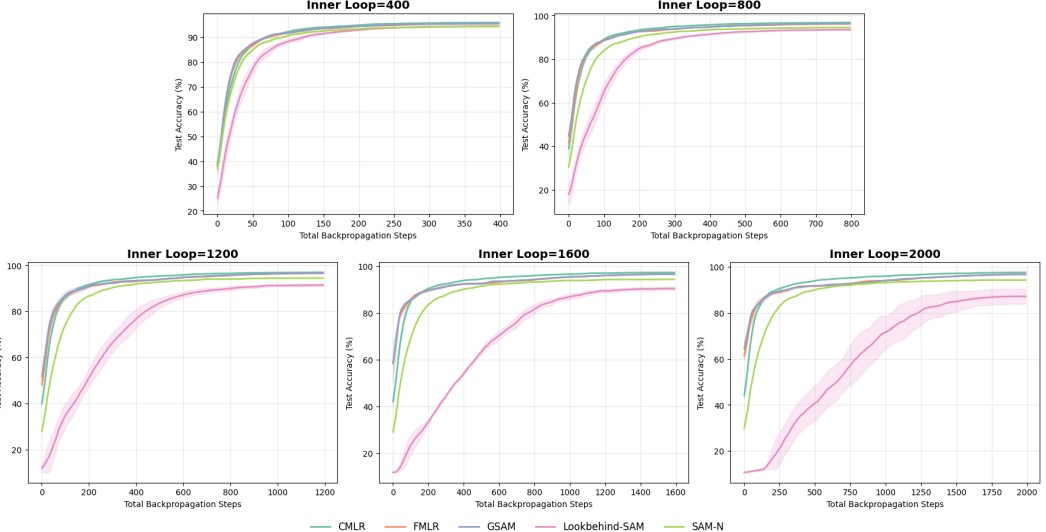

Figure 3: Test accuracy versus backpropagation steps on ResNet-34/CIFAR-10. We compare five algorithms under an equal number of backpropagation steps (400, 800, 1200, 1600, 2000) to provide a fair measure of computational cost.

## 5.2 VISION TRANSFORMERS

We evaluated ViT-Tiny, ViT-Small, and ViT-Base models on both CIFAR-10 and CIFAR-100, with all results averaged over three independent runs to ensure reproducibility. To align with the optimal results reported in contemporary works (Dosovitskiy et al., 2020; Zhao et al., 2024; Yun, 2025), models were trained from scratch for 300 epochs using the AdamW optimizer ($\beta_1 = 0.9, \beta_2 = 0.999$) and a cosine annealing scheduler with an initial learning rate of 1e-3. Additionally,We applied a weight decay of $0.03$ and standard data augmentations (4-pixel padding, random crop-ping, and horizontal flipping). Finally, to align with prior experiments, the hyperparameters for our method, CMLR, were selected from the following sets $K = 10, \alpha \in [0.7, 1.0], \rho \in \{0.05, 0.1\}, \lambda \in [0.05, 0.15]$ to demonstrate its cross-architecture robustness. The result can be seen in Table 2.

## 5.3 TRANSFORMER ENCODER−BASED ARCHITECTURE

We evaluate all algorithms on eight natural language understanding tasks from the GLUE bench-mark (Wang et al., 2018), using a standard Transformer-based architecture (DistilBERT). These tasks cover a broad range of linguistic phenomena, including sentiment classification (SST-2), lin-guistic acceptability (CoLA), paraphrase detection (MRPC, QQP), semantic similarity (STS-B), and natural language inference (MNLI, QNLI, RTE). Following GLUE protocol, we report task-specific metrics: Matthews correlation for CoLA, F1 score for MRPC and QQP, Pearson correlation for STS-B, and accuracy for the remaining tasks. Finally, we report an aggregate GLUE score computed as the unweighted average over seven tasks, excluding STS-B.

We fine-tune each model using the AdamW optimizer with weight decay fixed at 0.01. For GSAM, Lookbehind-SAM, FMLR, and CMLR, we set the lookahead step: $K = 2$. Hyper-parameters are selected from the following ranges: $\rho \in \{0.001, 0.005, 0.01, 0.05, 0.1\}, \lambda \in \{0.001, 0.005, 0.01, 0.05, 0.1\}$. For all methods, we use the same training setup, except for learning rate, batch size, and number of epochs, which are tuned per task and detailed in Appendix F.

All experiments initialize the model from the publicly available pre-trained DistilBERT-base-uncased checkpoint, with a standard classification head or a regression output layer (for STS-B). Comprehensive results for all eight tasks are summarized in Table 3.

Table 2: Performance comparison of CMLR against baseline optimizers in ViT models (Test Accuracy %).

| Optimizer | CIFAR-10 (Test Accuracy %) | | |
| --- | --- | --- | --- |
| | ViT-Tiny | ViT-Small | ViT-Base |
| AdamW | $85.17_{\pm 0.15}$ | $85.93_{\pm 0.12}$ | $85.64_{\pm 0.16}$ |
| SAM | $85.86_{\pm 0.08}$ | $86.79_{\pm 0.09}$ | $86.82_{\pm 0.10}$ |
| CR-SAM | $86.08_{\pm 0.09}$ | $86.96_{\pm 0.11}$ | $87.00_{\pm 0.12}$ |
| Lookabehind-SAM | $86.36_{\pm 0.08}$ | $87.26_{\pm 0.08}$ | $87.32_{\pm 0.09}$ |
| GSAM | $86.69_{\pm 0.17}$ | $87.48_{\pm 0.08}$ | $87.58_{\pm 0.09}$ |
| FMLR | $86.54_{\pm 0.17}$ | $87.42_{\pm 0.09}$ | $87.47_{\pm 0.11}$ |
| **CMLR** | $\mathbf{86.95}_{\pm 0.07}$ | $\mathbf{87.81}_{\pm 0.09}$ | $\mathbf{88.30}_{\pm 0.08}$ |

| Optimizer | CIFAR-100 (Test Accuracy %) | | |
| --- | --- | --- | --- |
| | ViT-Tiny | ViT-Small | ViT-Base |
| AdamW | $58.87_{\pm 0.23}$ | $61.39_{\pm 0.21}$ | $61.75_{\pm 0.28}$ |
| SAM | $60.16_{\pm 0.16}$ | $62.15_{\pm 0.15}$ | $62.29_{\pm 0.20}$ |
| CR-SAM | $60.20_{\pm 0.17}$ | $62.35_{\pm 0.14}$ | $62.46_{\pm 0.21}$ |
| Lookbehind-SAM | $60.57_{\pm 0.16}$ | $62.63_{\pm 0.13}$ | $62.78_{\pm 0.19}$ |
| GSAM | $60.89_{\pm 0.14}$ | $62.91_{\pm 0.12}$ | $63.07_{\pm 0.18}$ |
| FMLR | $60.81_{\pm 0.25}$ | $62.84_{\pm 0.12}$ | $62.91_{\pm 0.18}$ |
| **CMLR** | $\mathbf{61.21}_{\pm 0.14}$ | $\mathbf{63.42}_{\pm 0.12}$ | $\mathbf{63.46}_{\pm 0.17}$ |

Table 3: Performance on GLUE tasks using DistilBERT. Best results per row are bolded. Metrics: MCC for CoLA, F1 for MRPC and QQP, Pearson for STS-B, Accuracy for others.

| Task | AdamW | SAM | CRSAM | GSAM | LookbehindSAM | FMLR | CMLR |
| --- | --- | --- | --- | --- | --- | --- | --- |
| CoLA | 56.69 | 57.69 | 58.29 | 59.03 | 58.79 | 58.97 | **59.19** |
| SST-2 | 91.28 | 92.08 | 92.78 | 93.53 | 93.38 | 93.48 | **93.58** |
| MRPC | 89.15 | 89.85 | 90.45 | 91.19 | 90.95 | 91.13 | **91.35** |
| STS-B | 86.99 | 88.19 | 88.89 | 89.63 | 89.39 | 89.57 | **89.69** |
| QQP | 86.85 | 87.85 | 88.45 | 89.19 | 88.95 | 89.13 | **89.35** |
| MNLI | 82.17 | 83.17 | 83.87 | 84.62 | 84.47 | 84.57 | **84.67** |
| QNLI | 88.87 | 90.17 | 90.77 | 91.51 | 91.27 | 91.45 | **91.67** |
| RTE | 61.73 | 63.23 | 63.93 | 64.68 | 64.53 | 64.63 | **64.73** |
| **Avg (GLUE)** | 79.53 | 80.58 | 81.22 | 81.96 | 81.77 | 81.91 | **82.08** |

## 6 CONCLUSION

This paper presented a principled solution to the instability paradox in multi-step sharpness-aware training, identifying perturbation gradient instability as the primary bottleneck and proposing CMLR, an alternative built on a more robust foundation that replaces unstable approximations with a more accurate central-difference scheme, embedded within an efficient, momentum Lookahead framework; extensive empirical results validate this design, showing CMLR produces more stable optimization trajectories and consistently outperforms existing gradient regularization methods on benchmarks, while comprehensive convergence and variance reduction analyses further underscore its theoretical rigor, with ablation studies confirming the critical roles of each component in achieving superior performance across diverse architectures including CNNs, Vision Transformers and Transformer encoder–based architecture.

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

# A   ALGORITHM

---

**Algorithm 2** CMLR (simplified version for convergence analysis)

---

**Require:** Same inputs as Algorithm 1.
 1: **for** $t = 0, 1, \ldots, T - 1$ **do**
 2:     Follow Algorithm 1 until the $(K - 2)$-th inner step.
 3:     Compute $d_{K-1}$ as in line 12 of Algorithm 1.
 4:     Synchronize fast and slow weights with Lookahead:

$$w_{t+1}, w_{t,K} \leftarrow w_t + \alpha\big(\mathcal{A}(w_{t,K-1}, \eta_{t,K-1}, d_{K-1}) - w_t\big).$$

 5: **end for**
 6: **return** $w_T$.
**Ensure:** CMLR trained model.

---

# B   DETAILED CONVERGENCE ANALYSIS FOR CLR

## B.1   NECESSARY ASSUMPTIONS

**Assumption B.1** (L-Smoothness of the Gradient). *The loss function, $\mathcal{L}(w)$, is $L_1$-smooth. And this condition is equivalent to its Hessian matrix having a bounded norm, i.e., $\|\nabla^2 \mathcal{L}(w)\| \leq L_1$.*

$$\|\nabla\mathcal{L}(w_1) - \nabla\mathcal{L}(w_2)\| \leq L_1 \|w_1 - w_2\| \tag{14}$$

**Assumption B.2** (Gradient Oracle Properties). *The stochastic gradient $g(w; \xi)$ is an unbiased estimator of the true gradient and has its variance bounded by $\sigma^2$.*

$$\mathbb{E}_\xi[g(w; \xi)] = \nabla\mathcal{L}(w), \quad \mathbb{E}_\xi[\|g(w; \xi) - \nabla\mathcal{L}(w)\|^2] \leq \sigma^2 \tag{15}$$

**Assumption B.3** (Lipschitz Continuous Hessian). *The Hessian $\nabla^2\mathcal{L}(w)$ is $L_2$-Lipschitz continuous.*

$$\|\nabla^2\mathcal{L}(w_1) - \nabla^2\mathcal{L}(w_2)\| \leq L_2 \|w_1 - w_2\| \tag{16}$$

**Assumption B.4** (Bounded Fourth Moment). *The stochastic gradient has a bounded fourth moment. $\exists M > 0$ such that*

$$\mathbb{E}\|g(w; \xi) - \nabla\mathcal{L}(w)\|^4 \leq M \tag{17}$$

## B.2   PROOF OF RELAXED DESCENT LEMMA

**Lemma B.5** (Relaxed Descent Lemma). *Under Assumptions 1 and 3, the sequence of iterates $\{\hat{y}_s\}$ generated by the GLEAP algorithm satisfies the following inequality:*

$$F(\hat{y}_{s+1}) \leq F(\hat{y}_s) - \alpha\eta_s\langle\nabla F(\hat{y}_s), d(y_s; \xi_s)\rangle + \frac{L_1\alpha^2\eta_s^2}{2}\|d(y_s; \xi_s)\|^2 + 2\lambda L_1\alpha\eta_s\|d(y_s; \xi_s)\| \tag{18}$$

*Proof.* The proof starts from the fundamental theorem of calculus applied to $F$:

$$F(y) - F(x) = \int_0^1 \langle\nabla F(x + t(y - x)), y - x\rangle \, dt$$

$$= \langle\nabla F(x), y - x\rangle + \int_0^1 \langle\nabla F(x + t(y - x)) - \nabla F(x), y - x\rangle \, dt \tag{19}$$

We bound the integrand using the Cauchy-Schwarz inequality and the relaxed smoothness property:

$$\langle\nabla F(x + t(y - x)) - \nabla F(x), y - x\rangle \leq \|\nabla F(x + t(y - x)) - \nabla F(x)\| \cdot \|y - x\|$$

$$\leq \|\nabla\mathcal{L}(x + t(y - x)) - \nabla\mathcal{L}(x)\| \cdot \|y - x\|$$

$$+ \lambda(\|\nabla\|\nabla\mathcal{L}(x + t(y - x))\|\| + \lambda\|\nabla\|\nabla\mathcal{L}(x)\|\|) \cdot \|y - x\|$$

$$\leq L_1 t\|y - x\|^2 + 2\lambda L_1\|y - x\| \tag{20}$$

Substituting this back into the integral:

$$F(y) - F(x) \leq \langle\nabla F(x), y - x\rangle + \int_0^1 (L_1 t\|y - x\|^2 + 2\lambda L_1\|y - x\|)dt$$

$$= \langle\nabla F(x), y - x\rangle + \frac{L_1}{2}\|y - x\|^2 + 2\lambda L_1\|y - x\| \tag{21}$$

Now, we let $x = \hat{y}_s$ and $y = \hat{y}_{s+1}$. The update step is $\hat{y}_{s+1} - \hat{y}_s = -\alpha\eta_s d(y_s; \xi_s)$:

$$F(\hat{y}_{s+1}) \leq F(\hat{y}_s) + \langle\nabla F(\hat{y}_s), -\alpha\eta_s d(y_s; \xi_s)\rangle + \frac{L_1}{2}\|\alpha\eta_s d(y_s; \xi_s)\|^2 + 2\lambda L_1\|\alpha\eta_s d(y_s; \xi_s)\|$$

$$= F(\hat{y}_s) - \alpha\eta_s\langle\nabla F(\hat{y}_s), d(y_s; \xi_s)\rangle + \frac{L_1\alpha^2\eta_s^2}{2}\|d(y_s; \xi_s)\|^2 + 2\lambda L_1\alpha\eta_s\|d(y_s; \xi_s)\| \tag{22}$$

This completes the proof of the lemma. $\qquad\square$

### B.3 PROOF OF MAIN CONVERGENCE THEOREM

**Theorem B.6** (Convergence of CLR for Non-Convex Objectives). *Under Assumptions B.1-B.3, if the learning rate $\eta_s$ is sufficiently small and satisfies $\sum_{s=0}^{\infty}\eta_s = \infty$ and $\sum_{s=0}^{\infty}\eta_s^p < \infty$ for $p \geq 2$, the iterates sequence generated by CLR satisfy:*

$$\liminf_{s\to\infty}\mathbb{E}[\|\nabla F(\hat{y}_s)\|^2] = \mathcal{O}(\lambda). \tag{23}$$

*Proof.* The proof begins by taking the total expectation of Equation 18. We must bound the expectation of the three terms on the right-hand side.

First, we bound the inner product term: Let $B(y_s) = \mathbb{E}[d(y_s, \xi_s)] - \nabla F(y_s)$.

$$\mathbb{E}[\langle\nabla F(\hat{y}_s), d(y_s; \xi_s)\rangle] = \mathbb{E}[\langle\nabla F(\hat{y}_s), \nabla F(y_s)\rangle] + \mathbb{E}[\langle\nabla F(\hat{y}_s), B(y_s)\rangle] \tag{24}$$

The first term of RHS:

$$\mathbb{E}[\langle\nabla F(\hat{y}_s), \nabla F(y_s)\rangle] = \frac{1}{2}\mathbb{E}[\|\nabla F(\hat{y}_s)\|^2] + \frac{1}{2}\mathbb{E}[\|\nabla F(y_s)\|^2] - \frac{1}{2}\mathbb{E}[\|\nabla F(\hat{y}_s) - \nabla F(y_s)\|^2]$$

$$\geq \frac{1}{2}\mathbb{E}[\|\nabla F(\hat{y}_s)\|^2] + \frac{1}{2}\mathbb{E}[\|\nabla F(y_s)\|^2] - L_1^2\mathbb{E}[\|\hat{y}_s - y_s\|^2] - 2\lambda L_1 \tag{25}$$

where

$$\mathbb{E}[\|\nabla F(\hat{y}_s) - \nabla F(y_s)\|^2] \leq 2\mathbb{E}[\|\nabla L(\hat{y}_s) - \nabla L(y_s)\|^2] + 2\lambda(\mathbb{E}[\|\nabla^2 L(\hat{y}_s)\frac{\nabla L(\hat{y}_s)}{\|\nabla L(\hat{y}_s)\| + \varepsilon}\|^2]$$

$$+ \mathbb{E}[\|\nabla^2 L(y_s)\frac{\nabla L(y_s)}{\|\nabla L(y_s)\| + \varepsilon}\|^2])$$

$$= 2L_1^2\mathbb{E}[\|\hat{y}_s - y_s\|^2] + 4\lambda L_1. \tag{26}$$

The second term of RHS:

$$\mathbb{E}[\langle\nabla F(\hat{y}_s), B(y_s)\rangle] \geq -\mathbb{E}[\|\nabla F\hat{y}_s)\|\|B(y_s)\|]$$

$$\geq -\frac{\varepsilon_1}{2}\mathbb{E}[\|\nabla F(\hat{y}_s)\|^2 - \frac{1}{2\varepsilon_1}\mathbb{E}[\|B(y_s)\|^2]] \tag{27}$$

where

$$\mathbb{E}[\|B(y_s)\|]$$

$$= \lambda\mathbb{E}[\|\frac{1}{2\rho}(\nabla L(y_s + \rho\frac{\nabla L(y_s; \xi_s)}{\|\nabla L(y_s; \xi_s)\| + \varepsilon}) - \nabla L(y_s - \rho\frac{\nabla L(y_s; \xi_s)}{\|\nabla L(y_s; \xi_s)\| + \varepsilon})) - \nabla^2 L(y_s)\frac{\nabla L(y_s)}{\|\nabla L(y_s)\| + \varepsilon}\|]$$

$$\leq \lambda\mathbb{E}[\|\frac{1}{2\rho}(\nabla L(y_s + \rho\frac{\nabla L(y_s; \xi_s)}{\|\nabla L(y_s; \xi_s)\| + \varepsilon}) - \nabla L(y_s - \rho\frac{\nabla L(y_s; \xi_s)}{\|\nabla L(y_s; \xi_s)\| + \varepsilon})) - \nabla^2 L(y_s; \xi_s)\frac{\nabla L(y_s; \xi_s)}{\|\nabla L(y_s; \xi_s)\| + \varepsilon}\|]$$

$$+ \lambda\mathbb{E}[\|\nabla^2 L(y_s; \xi_s)\frac{\nabla L(y_s; \xi_s)}{\|\nabla L(y_s; \xi_s)\| + \varepsilon} - \nabla^2 L(y_s)\frac{\nabla L(y_s)}{\|\nabla L(y_s)\| + \varepsilon}\|]$$

$$\leq \lambda\frac{L_2\rho^2}{6} + 2\lambda L_1 \tag{28}$$

Thus, we have:

$$\mathbb{E}[\langle\nabla F(\hat{y}_s), d(y_s; \xi_s)\rangle] \geq \frac{1}{2}(1 - \varepsilon_1)\mathbb{E}[\|\nabla F(\hat{y}_s)\|^2] + \frac{1}{2}\mathbb{E}[\|\nabla F(y_s)\|^2]$$

$$- L_1^2(1 - \alpha)^2\mathbb{E}[\|y_s - y_{\lfloor s/K\rfloor K}\|^2] - C_1. \tag{29}$$

where $C_1 = \frac{\lambda^2 L_2^2 \rho^4}{18\varepsilon_1} + \frac{2\lambda^2 L_1^2}{\varepsilon_1} + 2\lambda L_1$.

Second, the squared norm of the update direction is bounded:

$$\mathbb{E}[\|d(y_s;\xi_s)\|^2] = \mathbb{E}\left[\mathrm{Var}_{\xi_s}(d(y_s;\xi_s)) + \|\mathbb{E}_{\xi_s}[d(y_s;\xi_s)]\|^2\right] \tag{30}$$

To bound the variance, let $\delta_s = g(y_s;\xi_s) - \nabla\mathcal{L}(y_s)$.

$$\mathbb{E}\left[\mathrm{Var}_{\xi_s}(d(y_s;\xi_s))\right] = \mathbb{E}\left[\mathbb{E}_{\xi_s}\left[\left\|\delta_s + \frac{\lambda}{2\rho}(\delta_s^+ - \delta_s^-)\right\|^2\right]\right]$$

$$\leq \mathbb{E}\left[\mathbb{E}_{\xi_s}\left[2\|\delta_s\|^2 + 2\left\|\frac{\lambda}{2\rho}(\delta_s^+ - \delta_s^-)\right\|^2\right]\right]$$

$$\leq \mathbb{E}\left[\mathbb{E}_{\xi_s}\left[2\|\delta_s\|^2 + \frac{\lambda^2}{\rho^2}\left(\|\delta_s^+\|^2 + \|\delta_s^-\|^2\right)\right]\right]$$

$$\leq \mathbb{E}\left[2\sigma^2 + \frac{\lambda^2}{\rho^2}(\sigma^2 + \sigma^2)\right] = 2\sigma^2\left(1 + \frac{\lambda^2}{\rho^2}\right) \tag{31}$$

Thus, we have

$$\mathbb{E}[\|d(y_s;\xi_s)\|^2] \leq 2\sigma^2\left(1 + \frac{\lambda^2}{\rho^2}\right) + \mathbb{E}\left[2\|\nabla F(y_s)\|^2 + 2\|B(y_s)\|^2\right]$$

$$= 2\mathbb{E}\left[\|\nabla F(y_s)\|^2\right] + C_2 \tag{32}$$

where $C_2 = 2\sigma^2\left(1 + \frac{\lambda^2}{\rho^2}\right) + 2\lambda^2(\frac{L_2\rho^2}{6} + 2L_1)^2$.

Third, we handle the linear norm term using Jensen's inequality and AM-GM:

$$\mathbb{E}[\|d(y_s;\xi_s)\|] \leq \sqrt{\mathbb{E}[\|d(y_s;\xi_s)\|^2]}$$

$$\leq \sqrt{C_2 + 2\mathbb{E}[\|\nabla F(y_s)\|^2]}$$

$$\leq \frac{\varepsilon_2}{2} + \frac{C_2 + 2\mathbb{E}[\|\nabla F(y_s)\|^2]}{2\varepsilon_2} \tag{33}$$

Substituting these bounds back into the expectation of Equation 18 and summing over one outer loop ($k = 0, \ldots, K-1$) with a fixed learning rate $\eta_{tK}$, we get the main recurrence relation:

$$\mathbb{E}[F(\hat{y}_{(t+1)K})] - \mathbb{E}[F(\hat{y}_{tK})] \leq -\frac{\alpha\eta_{tK}}{2}(1 - \varepsilon_1)\sum_{k=0}^{K-1}\mathbb{E}[\|\nabla F(\hat{y}_{tK+k})\|^2]$$

$$- \left(\frac{\alpha\eta_{tK}}{2} - L_1\alpha^2\eta_{tK}^2 - \frac{2\lambda L_1\alpha\eta_{tK}}{\varepsilon_2}\right)\sum_{k=0}^{K-1}\mathbb{E}[\|\nabla F(y_{tK+k})\|^2]$$

$$+ L_1^2(1-\alpha)^2\alpha\eta_{tK}\sum_{k=0}^{K-1}\mathbb{E}[\|y_{tK+k} - y_{tK}\|^2] + C_3 K \tag{34}$$

where $C_3$ collects constant error terms. The third term of RHS in Equation 34:

$$\mathbb{E}\left[\|y_{tK+k} - y_{tK}\|^2\right] = \mathbb{E}\left[\left\|\sum_{j=0}^{k-1}(-\eta_{tK}d_{tK+j})\right\|^2\right]$$

$$\leq \eta_{tK}^2\mathbb{E}\left[k\sum_{j=0}^{k-1}\|d_{tK+j}\|^2\right]$$

$$\leq \eta_{tK}^2 k\sum_{j=0}^{k-1}\left(C_2 + 2\mathbb{E}\left[\|\nabla F(y_{tK+j})\|^2\right]\right)$$

$$= k^2\eta_{tK}^2 C_2 + 2k\eta_{tK}^2\sum_{j=0}^{k-1}\mathbb{E}\left[\|\nabla F(y_{tK+j})\|^2\right] \tag{35}$$

Summing over $k$:

$$\sum_{k=0}^{K-1} \mathbb{E}\left[\|y_{tK+k} - y_{tK}\|^2\right] \leq C_2 \eta_{tK}^2 \sum_{k=0}^{K-1} k^2 + 2\eta_{tK}^2 \sum_{k=1}^{K-1} k \left(\sum_{j=0}^{K-1} \mathbb{E}\left[\|\nabla F(y_{tK+j})\|^2\right]\right)$$

$$= \underbrace{C_2 \frac{(K-1)K(2K-1)}{6}}_{C_4} \eta_{tK}^2 + \underbrace{K(K-1)}_{C_5} \eta_{tK}^2 \sum_{j=0}^{K-1} \mathbb{E}\left[\|\nabla F(y_{tK+j})\|^2\right] \tag{36}$$

$$= C_4 \eta_{tK}^2 + C_5 \eta_{tK}^2 \sum_{k=0}^{K-1} \mathbb{E}[\|\nabla F(y_{tK+k})\|^2]$$

Next, we bound the second term of RHS in Equation 34:

$$\sum_{k=0}^{K-1} E\left[\|\nabla F(y_{tK+k})\|^2\right] \leq \sum_{k=0}^{K-1} 2\mathbb{E}\left[\|\nabla F(\hat{y}_{tK+k})\|^2\right] + 2\mathbb{E}\left[\|\nabla F(y_{tK+k}) - \nabla F(\hat{y}_{tK+k})\|^2\right]$$

$$\leq \sum_{k=0}^{K-1} 2\mathbb{E}\left[\|\nabla F(\hat{y}_{tK+k})\|^2\right] + 4L_1^2 \mathbb{E}\left[\|y_{tK+k} - \hat{y}_{tK+k}\|^2\right] + 8\lambda^2 L_1^2$$

$$= 2\sum_{k=0}^{K-1} \mathbb{E}\left[\|\nabla F(\hat{y}_{tK+k})\|^2\right] + 4L_1^2(1-\alpha)^2 \sum_{k=0}^{K-1} \mathbb{E}\left[\|y_{tK+k} - y_{tK}\|^2\right] + 8K\lambda^2 L_1^2 \tag{37}$$

By substituting into equation Equation 36, we can obtain:

$$\sum_{k=0}^{K-1} \mathbb{E}\left[\|\nabla F(y_{tK+k})\|^2\right] \leq 2\sum_{k=0}^{K-1} \mathbb{E}\left[\|\nabla F(\hat{y}_{tK+k})\|^2\right] + 8K\lambda^2 L_1^2$$

$$+ 4L_1^2(1-\alpha)^2 \left(C_4 \eta_{tK}^2 + C_5 \eta_{tK}^2 \sum_{k=0}^{K-1} \mathbb{E}[\|\nabla F(y_{tK+k})\|^2]\right) \tag{38}$$

$$\implies \left(1 - 4L_1^2(1-\alpha)^2 C_5 \eta_{tK}^2\right) \sum_{k=0}^{K-1} \mathbb{E}\left[\|\nabla F(y_{tK+k})\|^2\right] \leq 2\sum_{k=0}^{K-1} \mathbb{E}\left[\|\nabla F(\hat{y}_{tK+k})\|^2\right]$$

$$+ 4L_1^2(1-\alpha)^2 C_4 \eta_{tK}^2 + 8K\lambda^2 L_1^2 \tag{39}$$

$$\implies \sum_{k=0}^{K-1} \mathbb{E}\left[\|\nabla F(y_{tK+k})\|^2\right] \leq C_6 \sum_{k=0}^{K-1} \mathbb{E}\left[\|\nabla F(\hat{y}_{tK+k})\|^2\right] + C_7. \tag{40}$$

where $C_6, C_7$ are constants for a sufficiently small $\eta_{tK}$.

Substituting these bounds back into the main recurrence Equation 34 allows us to eliminate all dependencies on the fast weights $y_{tK+k}$. After collecting terms, we arrive at the simplified one-step recurrence for the interpolated weights $\hat{y}$:

$$\mathbb{E}[F(\hat{y}_{(t+1)K})] - \mathbb{E}[F(\hat{y}_{tK})] \leq -C_8 \sum_{k=0}^{K-1} \mathbb{E}[\|\nabla F(\hat{y}_{tK+k})\|^2] + C_9 \tag{41}$$

where

$$C_8 = \frac{\alpha\eta}{2}(1 - \varepsilon_1) + C_6 \left(\frac{\alpha\eta}{2} - L_1\alpha^2\eta^2 - \frac{2\lambda L_1\alpha\eta}{\varepsilon_2} - L_1^2(1-\alpha)^2\alpha C_5\eta^3\right) \tag{42}$$

is a positive coefficient, and

$$C_9 = C_3 K + L_1^2(1-\alpha)^2\alpha C_4\eta^3 - C_7 \left(\frac{\alpha\eta}{2} - L_1\alpha^2\eta^2 - \frac{2\lambda L_1\alpha\eta}{\varepsilon_2} - L_1^2(1-\alpha)^2\alpha C_5\eta^3\right) \tag{43}$$

is a term collecting various constants and error terms.

To analyze the long-term behavior, we sum the inequality from $t = 0$ to $T - 1$:

$$C_8 \sum_{l=0}^{\tau-1} \mathbb{E}[\|\nabla F(\hat{y}_{t\tau+l})\|^2] \leq \mathbb{E}[F(\hat{y}_{t\tau})] - \mathbb{E}[F(\hat{y}_{(t+1)\tau})] + C_9$$

$$\implies \sum_{t=0}^{T-1} C_8 \sum_{l=0}^{\tau-1} \mathbb{E}[\|\nabla F(\hat{y}_{t\tau+l})\|^2] \leq \sum_{t=0}^{T-1} \left( \mathbb{E}[F(\hat{y}_{t\tau})] - \mathbb{E}[F(\hat{y}_{(t+1)\tau})] \right) + \sum_{t=0}^{T-1} C_9 \quad (44)$$

The first term on the right-hand side is a telescoping series, bounded by $\mathbb{E}[F(\hat{y}_0)] - F_{\inf}$. The sum of $C_9$ contains terms proportional to various powers of the learning rate and $\lambda$.

$$\sum_{s=0}^{TK-1} O(\eta_s) \mathbb{E}[\|\nabla F(\hat{y}_s)\|^2] \leq F(\hat{y}_0) - F_{\inf} + O(\lambda) \sum_{s=0}^{TK-1} \eta_s + O(1) \sum_{s=0}^{TK-1} (\eta_s^2 + \cdots + \eta_s^5) \quad (45)$$

Dividing by $\sum_{s=0}^{TK-1} \eta_s$ and taking the limit $T \to \infty$, we leverage the learning rate conditions: $\sum \eta_s = \infty$ and $\sum \eta_s^p < \infty$ for $p \geq 2$. Under these conditions, the terms $(F(\hat{y}_0) - F_{\inf})/\sum \eta_s$ and $(\sum \eta_s^2)/\sum \eta_s$ both converge to zero. The dominant non-vanishing term on the right-hand side is therefore proportional to $\lambda$. This leads to the conclusion:

$$\liminf_{T \to \infty} \frac{\sum_{s=0}^{TK-1} \eta_s \mathbb{E}[\|\nabla F(\hat{y}_s)\|^2]}{\sum_{s=0}^{TK-1} \eta_s} \leq \mathcal{O}(\lambda) \quad (46)$$

This result confirms that CLR converges to a neighborhood of a stationary point of the regularized objective $F(w)$, with the size of this neighborhood governed by $\lambda$. $\qquad\square$

## C   DETAILED CONVERGENCE ANALYSIS FOR CMLR

For the sake of convenience, we provide some new variable definitions and supplementary explanations before describing the theorem. In the gradient normalization step of the algorithm implementation, in order to avoid the denominator being 0, we add a small perturbation $S_\varepsilon(z) \triangleq \frac{z}{\|z\| + \varepsilon}$ to replace $\frac{z}{\|z\|}$ in practical implementation. Moreover, we give other definitions following: $\hat{v}_k = S_\varepsilon(v_k), u_k = S_\varepsilon(\nabla\mathcal{L}(y_k)), \tilde{g}_k \triangleq \frac{1}{2}\left( \frac{g_k^+}{\|g_k^+\| + \varepsilon} + \frac{g_k^-}{\|g_k^-\| + \varepsilon} \right)$

**Theorem 4.1** (Convergence of CMLR for Non-Convex Objectives). *Under the same conditions in Theorem B.6, the iterative sequence generated by CMLR satisfies:*

$$\liminf_{s \to \infty} \mathbb{E}[\|\nabla F(\hat{y}_s)\|^2] = \mathcal{O}(\lambda). \quad (11)$$

*Proof.* Re estimate of $B(y_s)$ in Equation 28.

$$B(y_s) = \lambda \left( \mathbb{E}\left[ \frac{\nabla\mathcal{L}(y_s + \rho\hat{v}_s) - \nabla\mathcal{L}(y_s - \rho\hat{v}_s)}{2\rho} \right] - \nabla^2\mathcal{L}(y_s) S_\varepsilon(\nabla\mathcal{L}(y_s)) \right)$$

$$= \lambda(\mathbb{E}\left[ \frac{\nabla\mathcal{L}(y_s + \rho\hat{v}_s) - \nabla\mathcal{L}(y_s - \rho\hat{v}_s)}{2\rho} \right] - \nabla^2\mathcal{L}(y_s)\hat{v}_s)$$

$$+ \lambda(\nabla^2\mathcal{L}(y_s)\hat{v}_s - \nabla^2\mathcal{L}(y_s)\hat{S}_\varepsilon(\nabla\mathcal{L}(y_s)))$$

$$= \lambda(T_1 + T_2) \quad (47)$$

Bound for $T_1$. By the third-order Taylor expansion (or Hessian-Lipschitz control) the central-difference truncation is bounded pointwise by

$$\|\frac{\nabla\mathcal{L}(y_s + \rho\hat{v}_s) - \nabla\mathcal{L}(y_s - \rho\hat{v}_s)}{2\rho} - \nabla^2\mathcal{L}(y_s)\hat{v}_s\| \leq \frac{L_2\rho^2}{6}$$

for any unit vector $v$. Taking expectation yields

$$\|T_1\| \leq \frac{L_2\rho^2}{6}. \quad (48)$$

Bound for $T_2$. We first consider that $S_\varepsilon$ is $\max\{1, 1/\varepsilon\}$-Lipschitz.

$$S_\varepsilon(x) - S_\varepsilon(y) = \int_0^1 DS_\varepsilon\big(y + t(x - y)\big)(x - y)\, dt,$$

so that

$$\|S_\varepsilon(x) - S_\varepsilon(y)\| \leq \sup_{z \in \mathbb{R}^d} \|DS_\varepsilon(z)\|\, \|x - y\|.$$

A direct calculation gives

$$DS_\varepsilon(z) = \frac{1}{\|z\| + \varepsilon} I - \frac{zz^\top}{(\|z\| + \varepsilon)\|z\|},$$

hence

$$\|DS_\varepsilon(z)\| \leq \frac{1 + \|z\|}{\|z\| + \varepsilon}.$$

Taking the supremum over $\|z\| \geq 0$ yields

$$\|S_\varepsilon(x) - S_\varepsilon(y)\| \leq \max\left\{1, \frac{1}{\varepsilon}\right\} \|x - y\| = L_f \|x - y\|.$$

where $L_f \triangleq \max\{1, \frac{1}{\varepsilon}\}$. Applying this with $x = \hat{v}_s$ and $y = \nabla\mathcal{L}(y_s)$ and taking expectation yields

$$\big\|\mathbb{E}[\hat{v}_s] - S_\varepsilon(\nabla\mathcal{L}(y_s))\big\| \leq \mathbb{E}\|\hat{v}_s - S_\varepsilon(\nabla\mathcal{L}(y_s))\| \leq L_f(\varepsilon)\, \mathbb{E}\|e_s\|. \tag{49}$$

where $e_s \triangleq \hat{v}_s - u_s$. Then we have:

$$\|T_2\| \leq L_1\, L_f(\varepsilon)\, \mathbb{E}\|e_s\|, \tag{50}$$

Considering $v_{s+1} = \beta_s \hat{v}_s + (1 - \beta_s)\tilde{g}_s$, we have:

$$e_{s+1} = S_\varepsilon(\beta_s \hat{v}_s + (1 - \beta_s)\tilde{g}_s) - u_{s+1}$$
$$= \Big(S_\varepsilon(\beta_s \hat{v}_s + (1 - \beta_s)\tilde{g}_s) - S_\varepsilon(b_s)\Big) + \Big(S_\varepsilon(b_s) - u_{s+1}\Big), \tag{51}$$

where $b_s \triangleq \beta_s u_s + (1 - \beta_s)\bar{u}_s$ and $\bar{u}_s \triangleq \mathbb{E}[\tilde{g}_s]$. Using the Lipschitz property of $S_\varepsilon$,

$$\mathbb{E}\|S_\varepsilon(\beta_s \hat{v}_s + (1 - \beta_s)\tilde{g}_s) - S_\varepsilon(b_s)\|$$
$$\leq L_f\Big(\beta_s \mathbb{E}\|e_s\| + (1 - \beta_s)\mathbb{E}\|\tilde{g}_s - \bar{u}_s\|\Big),$$
$$\leq L_f(\beta_s \mathbb{E}\|e_s\| + (1 - \beta_s)\sqrt{\mathbb{E}\big\|\tilde{g}_s - \mathbb{E}[\tilde{g}_s]\big\|^2})$$
$$\leq L_f(\beta_s \mathbb{E}\|e_s\| + (1 - \beta_s)\sqrt{\frac{1}{2}\mathbb{E}\big\|S_\varepsilon(g_s^+) - S_\varepsilon(\nabla\mathcal{L}(y_s))\big\|^2 + \frac{1}{2}\mathbb{E}\big\|S_\varepsilon(g_s^-) - S_\varepsilon(\nabla\mathcal{L}(y_s))\big\|^2})$$
$$\leq L_f(\beta_s \mathbb{E}\|e_s\| + (1 - \beta_s)\sqrt{L_f^2 \mathbb{E}\|g_s^\pm - \nabla\mathcal{L}(y_s)\|^2})$$
$$\leq L_f(\beta_s \mathbb{E}\|e_s\| + (1 - \beta_s)L_f \frac{\sigma}{\sqrt{b}}) \tag{52}$$

$$\mathbb{E}\|S_\varepsilon(b_s) - u_{s+1}\| \leq L_f(\beta_s \|u_s - \nabla\mathcal{L}(y_{s+1})\| + (1 - \beta_s)\|\bar{u}_s - \nabla\mathcal{L}(y_{s+1})\|)$$
$$\leq L_f \beta_s(\|u_s - \nabla\mathcal{L}(y_s)\| + \|\nabla\mathcal{L}(y_s) - \nabla\mathcal{L}(y_{s+1})\|) + L_f(1 - \beta_s)(\|\bar{u}_s - u_s\| + \|u_s - \nabla\mathcal{L}(y_{s+1})\|)$$
$$\leq L_f \beta_s(\|\frac{\varepsilon}{\|\nabla\mathcal{L}(y_s)\| + \varepsilon}\nabla\mathcal{L}(y_s)\| + L_1\eta\|d_s\|) + L_f(1 - \beta_s)(L_f\frac{\sigma}{\sqrt{b}} + \|u_s - \nabla\mathcal{L}(y_{s+1})\|)$$
$$\leq L_f \beta_s(\varepsilon + L_1\eta\|d_s\|) + L_f(1 - \beta_s)(L_f\frac{\sigma}{\sqrt{b}} + \varepsilon + L_1\eta\|d_s\|)$$
$$\leq L_f\big(\varepsilon + L_1\eta\,\mathbb{E}\|d_s\| + (1 - \beta_s)L_f\frac{\sigma}{\sqrt{b}}\big). \tag{53}$$

Combining the two bounds yields the recursion

$$\mathbb{E}\|e_{s+1}\| \leq L_f \beta_s\, \mathbb{E}\|e_s\| + L_f\varepsilon + L_f L_1\eta\, \mathbb{E}\|d_s\| + 2L_f^2(1 - \beta_s)\frac{\sigma}{\sqrt{b}}. \tag{54}$$

$$\mathbb{E}\|e_s\| \le L_f^K \beta_{K-1}^K \mathbb{E}\|e_0\| + \frac{1 - L_f^K \beta_{K-1}^K}{1 - L_f \beta_{K-1}}(L_f \varepsilon + L_f L_1 \eta \mathbb{E}\|d_s\| + 2L_f^2(1 - \beta_0)\frac{\sigma}{\sqrt{b}})$$

$$\le \frac{1 - L_f^K \beta_{K-1}^K}{1 - L_f \beta_{K-1}} L_f L_1 \eta \mathbb{E}\|d_s\| + \frac{1 - L_f^K \beta_{K-1}^K}{1 - L_f \beta_{K-1}}(L_f \varepsilon + 2L_f^2(1 - \beta_0)\frac{\sigma}{\sqrt{b}}) \qquad (55)$$

$$\mathbb{E}\|B(y_s)\| \le \lambda L_1 L_f \frac{1 - L_f^K \beta_{K-1}^K}{1 - L_f \beta_{K-1}} L_f L_1 \eta \mathbb{E}\|d_s\| + \lambda \frac{L_2 \rho^2}{6}$$

$$+ \lambda L_1 L_f \frac{1 - L_f^K \beta_{K-1}^K}{1 - L_f \beta_{K-1}}(L_f \varepsilon + 2L_f^2(1 - \beta_0)\frac{\sigma}{\sqrt{b}})$$

$$= C_{10}\mathbb{E}\|d_s\| + C_{11} \qquad (56)$$

$$\mathbb{E}\|B(y_s)\|^2 = \|\mathbb{E}B\|^2 + \|B - \mathbb{E}B\|^2$$

$$\le (2C_{10}^2 \mathbb{E}\|d_s\|^2 + 2C_{11}^2) + 2\lambda^2(\mathbb{E}\|\nabla^2(y_s)(\hat{v}_s - \mathbb{E}\hat{v}_s)\|^2 + \mathbb{E}\|T_1\|^2)$$

$$\le 2C_{10}^2 \mathbb{E}\|d_s\|^2 + 2C_{11}^2 + 2\lambda^2 L_1^2 \mathbb{E}\|\hat{v}_s - \mathbb{E}\hat{v}_s\|^2 + \frac{\lambda^2 L_2^2 \rho^4}{18} \qquad (57)$$

$$\mathbb{E}\|v_s - \mathbb{E}v_s\|^2 = \mathbb{E}\big[\mathrm{Var}(v_{s+1} \mid \mathcal{F}_s)\big] + \mathrm{Var}\big(\mathbb{E}[v_{s+1} \mid \mathcal{F}_s]\big) \qquad (58)$$

First term:

$$\mathbb{E}\big[\mathrm{Var}(v_{s+1} \mid \mathcal{F}_s)\big] = \mathbb{E}\big[\mathrm{Var}(\beta_s \hat{v}_s + (1 - \beta_s)\tilde{g}_s \mid \mathcal{F}_s)\big]$$

$$= (1 - \beta_s)^2 \mathbb{E}\big[\mathrm{Var}(\tilde{g}_s \mid \mathcal{F}_s)\big]$$

$$\le (1 - \beta_s)^2 \frac{L_f^2 \sigma^2}{2b} \qquad (59)$$

$$\qquad (60)$$

Second term:

$$\mathrm{Var}\big(\mathbb{E}[v_{s+1} \mid \mathcal{F}_s]\big) = \mathrm{Var}\big(\beta_s \hat{v}_s + (1 - \beta_s)\mathbb{E}[\tilde{g}_s \mid \mathcal{F}_s]\big)$$

$$= \beta_s^2 \mathrm{Var}(\hat{v}_s)$$

$$= \beta_s^2 \mathbb{E}\|\hat{v}_s - \mathbb{E}\hat{v}_s\|^2$$

$$\le \beta_s^2 L_f^2 \mathbb{E}\|v_s - \mathbb{E}v_s\|^2 \qquad (61)$$

Combining the two terms, let $V_s \triangleq \mathbb{E}\|v_s - \mathbb{E}v_s\|^2$.

$$V_{s+1} \le \beta_s^2 L_f^2 V_s + (1 - \beta_s)^2 \frac{L_f^2 \sigma^2}{2b} \qquad (62)$$

$$V_s \le \frac{(1 - \beta_0)^2 L_f^2 \sigma^2 (1 - \beta_{K-1}^{2K} L_f^{2K})}{2b(1 - \beta_{K-1}^2 L_f^2)} \qquad (63)$$

$$\mathbb{E}\|B(y_s)\|^2 \le 2C_{10}^2 \mathbb{E}\|d_s\|^2 + C_{12} \qquad (64)$$

where $C_{12} = 2C_{11}^2 + \lambda^2 L_1^2 (1 - \beta_0)^2 L_f^2 \sigma^2 (1 - \beta_{K-1}^{2K} L_f^{2K})/b(1 - \beta_{K-1}^2 L_f^2) + \frac{\lambda^2 L_2^2 \rho^4}{18}$ Similar to the derivation of Theorem B.6, we can obtain the following recursive equation similar to Equation 34:

$$\mathbb{E}[F(\hat{y}_{(t+1)K})] - \mathbb{E}[F(\hat{y}_{tK})] \le -\frac{\alpha \eta_{tK}}{2}(1 - \varepsilon_1) \sum_{k=0}^{K-1} \mathbb{E}[\|\nabla F(\hat{y}_{tK+k})\|^2]$$

$$- \left(\frac{\alpha \eta_{tK}}{2} - L_1 \alpha^2 \eta_{tK}^2 - \frac{2\lambda L_1 \alpha \eta_{tK}}{\varepsilon_2} - \frac{C_{10}^2 \alpha \eta_s}{\varepsilon_1}\right) \sum_{k=0}^{K-1} \mathbb{E}[\|\nabla F(y_{tK+k})\|^2] \qquad (65)$$

$$+ L_1^2(1 - \alpha)^2 \alpha \eta_{tK} \sum_{k=0}^{K-1} \mathbb{E}[\|y_{tK+k} - y_{tK}\|^2] + \tilde{C}_3 K$$

where $\tilde{C}_3 = \alpha\eta_{tK}\tilde{C}_1 + \frac{C_{10}^2}{\varepsilon_1}\alpha\eta_{tK}\tilde{C}_2 + 2\lambda L_1\alpha\eta_{tK}(\frac{\varepsilon_2}{2} + \frac{2}{\varepsilon_2}), \tilde{C}_2 = C_2, \tilde{C}_1 = 2\lambda L_1 + \frac{C_{12}}{2\varepsilon_1}$. Then we have:

$$\mathbb{E}[F(\hat{y}_{(t+1)K})] - \mathbb{E}[F(\hat{y}_{tK})] \leq -\tilde{C}_8 \sum_{k=0}^{K-1} \mathbb{E}[\|\nabla F(\hat{y}_{tK+k})\|^2] + \tilde{C}_9 \tag{66}$$

where

$$\tilde{C}_8 = \frac{\alpha\eta_{tK}}{2}(1 - \varepsilon_1) + (\frac{\alpha\eta_{tK}}{2} - L_1\alpha^2\eta_{tK}^2 - \frac{2\lambda L_1\alpha\eta_{tK}}{\varepsilon_2} - \frac{C_{10}^2\alpha\eta_{tK}}{\varepsilon_1})\tilde{C}_6 - \tilde{C}_5\eta_{tK}^2\tilde{C}_6, \tag{67}$$

$$\tilde{C}_9 = \frac{\alpha\eta_{tK}}{2}(1 - \varepsilon_1) - (\frac{\alpha\eta_{tK}}{2} - L_1\alpha^2\eta_{tK}^2 - \frac{2\lambda L_1\alpha\eta_{tK}}{\varepsilon_2} - \frac{C_{10}^2\alpha\eta_{tK}}{\varepsilon_1})\tilde{C}_7$$
$$+ L_1^2(1 - \alpha)^2\alpha\eta_{tK}\tilde{C}_5\eta_{tK}^2\tilde{C}_7 + L_1^2(1 - \alpha)^2\alpha\eta_{tK} + \tilde{C}_4\eta_{tK}^2 + \tilde{C}_3K. \tag{68}$$

and $\tilde{C}_4 = C_4, \tilde{C}_5 = C_5, \tilde{C}_6 = C_6, \tilde{C}_7 = C_7$. Similar to the derivation of Equation 45, we can obtain a rough upper bound conclusion that is approximately consistent with Equation 46.

□

# D   DETAILED DERIVATION OF NOISY QUADRATIC ANALYSIS FOR LR

Our key insight is that this regularized objective is mathematically equivalent to a standard quadratic model but governed by an regularized Hessian, $H' = H + \lambda H^2$, where $\lambda$ represents the regularization strength. This allows us to directly apply the variance analysis tools for stochastic gradient descent (SGD) and Lookahead. Following the analysis framework, the asymptotic variance of the inner optimizer (GR-SGD) converges to a fixed point, which we denote as $V_R^*$.

**Theorem D.1** (Variance Reduction with LR). *When applying the Lookahead optimizer to the gradient-regularized noisy quadratic model, the asymptotic variance of the slow weights, $V_{LR}^*$, converges to the following fixed point:*

$$V_R^* = \mathcal{A}_2^{-1}\eta^2(H')^2\Sigma^2 \tag{69}$$

$$V_{LR}^* = \frac{\alpha^2\mathcal{A}_{2k}}{\alpha^2\mathcal{A}_{2k} + 2\alpha(1 - \alpha)\mathcal{A}_k}V_R^* \tag{70}$$

*where $\mathcal{A}_k$ are defined as:*

$$\mathcal{A}_k = (I - (I - \eta H')^k) \tag{71}$$

*Here, $H' = H + \lambda H^2$ is the regularized Hessian, $\eta$ is the inner learning rate, $\alpha$ is the slow weights step size, and $k$ is the number of inner loop steps.*

*Proof.* We first introduce a gradient regularization term to this objective, creating a new objective $\hat{L}_R(x)$:

$$\hat{L}_R(x) = \hat{\mathcal{L}}(x) + \frac{1}{2}\lambda\|\nabla\hat{\mathcal{L}}(x)\|^2 \tag{72}$$

The gradient of the original loss is $\nabla\hat{\mathcal{L}}(x) = H(x - c)$. Substituting this into the equation and assuming $H$ is symmetric ($H^\top = H$), we get:

$$\hat{L}_R(x) = \frac{1}{2}(x - c)^\top H(x - c) + \frac{1}{2}\lambda(x - c)^\top H^2(x - c)$$
$$= \frac{1}{2}(x - c)^\top(H + \lambda H^2)(x - c) \tag{73}$$

This shows that our regularized objective is equivalent to a standard noisy quadratic model with an regularized Hessian, defined as $H' = H + \lambda H^2$.

And then, to simplify the derivation, let us first define the intermediate term $\mathcal{A}_k$ as:

$$\mathcal{A}_k = I - (I - \gamma H')^k \tag{74}$$

The variance dynamics for the inner optimizer (R-SGD) are given by Wu et al. (2018):

$$V[x^{(t+1)}] = (I - \gamma H')^2 V[x^{(t)}] + \gamma^2(H')^2\Sigma \tag{75}$$

To find the asymptotic variance fixed point, $V_{\mathrm{R}}^*$, we set $V[x^{(t+1)}] = V[x^{(t)}] = V_{\mathrm{R}}^*$:

$$V_{\mathrm{R}}^* = (I - \gamma H')^2 V_{\mathrm{R}}^* + \gamma^2 (H')^2 \Sigma \tag{76}$$

$$(I - (I - \gamma H')^2) V_{\mathrm{R}}^* = \gamma^2 (H')^2 \Sigma \tag{77}$$

Recognizing that the term on the left, $I - (I - \gamma H')^2$, is exactly $\mathcal{A}_2$ from our definition in Equation 74:

$$\mathcal{A}_2 V_{\mathrm{R}}^* = \gamma^2 (H')^2 \Sigma \tag{78}$$

$$V_{\mathrm{R}}^* = \mathcal{A}_2^{-1} \gamma^2 (H')^2 \Sigma \tag{79}$$

The dynamics for the Lookahead slow weights $\phi_t$ are given by Zhang et al. (2019):

$$V[\phi_{t+1}] = [I - \alpha \mathcal{A}_k]^2 V[\phi_t] + \alpha^2 \left( \sum_{i=0}^{k-1} (I - \gamma H')^{2i} \right) \gamma^2 (H')^2 \Sigma \tag{80}$$

Note that we have rewritten the first term using $\mathcal{A}_k$: $(1-\alpha)I + \alpha(I - \gamma H')^k = (1-\alpha)I + \alpha(I - \mathcal{A}_k) = I - \alpha \mathcal{A}_k$.

To solve for the fixed point $V_{\mathrm{LR}}^*$, we set $V[\phi_{t+1}] = V[\phi_t] = V_{\mathrm{LR}}^*$:

$$V_{\mathrm{LR}}^* = \frac{\alpha^2 \left( \sum_{i=0}^{k-1} (I - \gamma H')^{2i} \right) \gamma^2 (H')^2 \Sigma}{I - [I - \alpha \mathcal{A}_k]^2} \tag{81}$$

Using the geometric series identity, the summation can be expressed with our notation:

$$\sum_{i=0}^{k-1} \left( (I - \gamma H')^2 \right)^i = \frac{I - (I - \gamma H')^{2k}}{I - (I - \gamma H')^2} = \mathcal{A}_{2k} \mathcal{A}_2^{-1} \tag{82}$$

Substituting this and the expression for $V_{GR}^*$ back into the equation:

$$
\begin{aligned}
V_{\mathrm{LR}}^* &= \frac{\alpha^2 \mathcal{A}_{2k} \mathcal{A}_2^{-1} \left( \gamma^2 (H')^2 \Sigma \right)}{I - [I - \alpha \mathcal{A}_k]^2} \\
&= \frac{\alpha^2 \mathcal{A}_{2k}}{I - (I^2 - 2\alpha \mathcal{A}_k + \alpha^2 \mathcal{A}_k^2)} \cdot \left( \mathcal{A}_2^{-1} \gamma^2 (H')^2 \Sigma \right) \\
&= \frac{\alpha^2 \mathcal{A}_{2k}}{2\alpha \mathcal{A}_k - \alpha^2 \mathcal{A}_k^2} \cdot V_{GR}^*
\end{aligned} \tag{83}
$$

The denominator can be factored as $2\alpha \mathcal{A}_k - \alpha^2 \mathcal{A}_k^2 = \alpha \mathcal{A}_k (2I - \alpha \mathcal{A}_k)$. To match the desired final form, we return to the denominator manipulation from the original paper, but expressed with $\mathcal{A}_k$:

$$
\begin{aligned}
& I - \left[ (1-\alpha)I + \alpha(I - \gamma H')^k \right]^2 \\
&= \alpha^2 \left( I - (I - \gamma H')^{2k} \right) + 2\alpha(1-\alpha) \left( I - (I - \gamma H')^k \right) \\
&= \alpha^2 \mathcal{A}_{2k} + 2\alpha(1-\alpha) \mathcal{A}_k
\end{aligned} \tag{84}
$$

This gives the final, simplified expression as specified:

$$V_{\mathrm{LR}}^* = \frac{\alpha^2 \mathcal{A}_{2k}}{\alpha^2 \mathcal{A}_{2k} + 2\alpha(1-\alpha) \mathcal{A}_k} V_R^* \tag{85}$$

This completes the proof. $\qquad \square$

The ratio is a multiplicative factor that is strictly less than 1 for any $\alpha \in (0, 1)$ and $k \geq 1$. This rigorously demonstrates that our method reduces the asymptotic variance compared to the inner GR-SGD optimizer alone, which contributes to the improved stability and convergence we observe in practice. The analysis of LR variance reduction followed the classical noise-propagation framework as in prior works. In contrast, our CMLR analysis adopts a spectral decomposition approach: we expand dynamics along each eigen-direction of $H$, thereby isolating both the data-induced variance and the additional contribution from momentum accumulation. Here, we provide the spectral analysis theorem for CMLR variance reduction.

# E   DETAILED DERIVATION OF NOISY QUADRATIC ANALYSIS FOR CMLR

we first define $n(z) \triangleq \frac{z}{\|z\|}, s \triangleq \rho H \hat{v}$, then we obtain $H(w-c) \to u$, $H(w-c) + \rho H \hat{v} \to g^+$ and $H(w-c) - \rho H \hat{v} \to g^-$.

**Lemma E.1** (Second-Order Remainder Bound). *Let $u \neq 0$. If $\|s\| \leq \frac{1}{2}\|u\|$, then there exists a remainder term $R$ such that*

$$\frac{g^+}{\|g^+\|} + \frac{g^-}{\|g^-\|} = 2\frac{u}{\|u\|} + R, \tag{86}$$

*with the strict norm bound*

$$\|R\| = O(\rho^2) \tag{87}$$

*Proof.* We first use a second-order Taylor expansion for the normalized map $n(x)$ at the point $u$:

$$n(u \pm s) = n(u) + Dn_u(\pm s) + \frac{1}{2}D^2 n_{u+\theta_\pm(\pm s)}(\pm s, \pm s),$$

for some $\theta_\pm \in (0, 1)$, where $Dn_u(h) = \frac{1}{\|u\|}(I - n(u)n(u)^\top)h$ . The first-order terms cancel out exactly: $Dn_u(s) + Dn_u(-s) = 0$.

The remainder term $R$ is therefore composed of the second-order terms:

$$R = \frac{1}{2}\left(D^2 n_{u+\theta_+ s}(s, s) + D^2 n_{u-\theta_- s}(-s, -s)\right).$$

For any $z \neq 0$, the second Fréchet derivative has the general upper bound given by:

$$\|D^2 n_u(h, k)\| \leq \left\|\frac{u^\top k}{\|u\|^3}(I - nn^\top)h\right\| + \left\|\frac{1}{\|u\|}Dn_u(k)n^\top h\right\| + \left\|\frac{1}{\|u\|}n(Dn_u(k))^\top h\right\|$$

$$\leq \frac{|u^\top k|}{\|u\|^3}\|I - nn^\top\|_{op}\|h\| + \frac{1}{\|u\|}\|Dn_u(k)\|\|n^\top h\| + \frac{1}{\|u\|}\|n\|\|(Dn_u(k))^\top h\|$$

$$\leq \frac{\|u\|\|k\|}{\|u\|^3}(1)\|h\| + \frac{1}{\|u\|}\left(\frac{\|k\|}{\|u\|}\right)(\|n\|\|h\|) + \frac{1}{\|u\|}(1)(\|Dn_u(k)\|\|h\|)$$

$$\leq \frac{1}{\|u\|^2}\|k\|\|h\| + \frac{1}{\|u\|^2}\|k\|\|h\| + \frac{1}{\|u\|^2}\|k\|\|h\|$$

$$= \frac{3}{\|u\|^2}\|h\|\|k\|.$$

where the operator $I - n(u)n(u)^\top$ is a projection onto the orthogonal complement of $n(u)$, so its operator norm is equal to its largest eigenvalue, which implies $\|I - nn^\top\|_{op} = 1$. We also use $\|n(u)\| = 1$ and the bound $\|Dn_u(k)\| \leq \frac{1}{\|u\|}\|k\|$. Given the assumption $\|s\| \leq \frac{1}{2}\|u\|$, we have that: $\|u \pm \theta_\pm s\| \geq \|u\| - \|s\| \geq \frac{1}{2}\|u\|$. This allows us to bound each component of the remainder:

$$\frac{1}{2}\|D^2 n_{u+\theta_+ s}(s, s)\| \leq \frac{1}{2}\|D^2 n_{u+\theta_+ s}\|_{op}\|s\|^2$$

$$\leq \frac{1}{2} \cdot \frac{3}{\|u+\theta_+ s\|^2}\|s\|^2$$

$$\leq \frac{1}{2} \cdot \frac{3}{(\frac{1}{2}\|u\|)^2}\|s\|^2 = 6\frac{\|s\|^2}{\|u\|^2}.$$

Summing the bounds for the two terms via the triangle inequality yields the final result:

$$\|R\| \leq 12\frac{\|s\|^2}{\|u\|^2}.$$

Therefore, substituting $s = \rho H v$ and $\|\hat{v}\| = 1$, we obtain

$$\left\|\frac{g^+}{\|g^+\|} + \frac{g^-}{\|g^-\|} - 2\frac{H(w-c)}{\|H(w-c)\|}\right\| \leq 12\frac{\rho^2\|H\|^2}{\|H(w-c)\|^2}.$$

In other words, as long as $\|H(w-c)\|$ is bounded from below, this difference is $O(\rho^2)$.

$\square$

Now, we consider true gradient: $g_r^{\pm} = u + s + \xi_r^{\pm}$, where $\xi_r^{\pm}$ satisfies Assumption B.2.

**Theorem 4.2** (Variance Reduction with CMLR). *Fix an eigenpair $(q, \mu)$ of $H$, with scalar projections $x_{t,k} = q^{\top} w_{t,k}$, $c_{t,k} = q^{\top} c$, and $\hat{v}_{k,q} = q^{\top} \hat{v}_k$. Let $a \triangleq 1 - \eta\mu$, define $A_{\text{eff}} \triangleq (1 - \alpha) + \alpha a^K$, $B_{\text{eff}} \triangleq \alpha\eta\mu$. Under Assumption B.2, Assumption B.4, $\|\nabla\mathcal{L}\| \geq g_{\min}$ and conditon in Lemma E.1, the steady-state variance of CMLR satisfies*

$$v_{\text{CMLR}} \leq v_{\text{LR}}(1 + \sqrt{\tau})^2 + \mathcal{O}(\frac{\rho^4\|H\|^4 + M}{g_{\min}^4}) \tag{12}$$

*where*

$$v_{\text{LR}} = \frac{\alpha^2(\eta\mu)^2\sigma^2 \dfrac{1 - a^{2K}}{1 - a^2}}{1 - A_{\text{eff}}^2}, \qquad \tau = 2d\frac{\lambda^2(1 - \beta)}{g_{\min}^2(1 + \beta)}, \tag{13}$$

*Proof.* We present a concise derivation starting from the algorithmic updates (quadratic loss). For the quadratic model one has the exact identity (no approximation)

$$g_k^{\pm} = H(w_{t,k} - c) \pm \rho H\hat{v}_k, \tag{88}$$

and therefore (using the coefficients $\frac{\rho+\lambda}{2\rho}, \frac{\rho-\lambda}{2\rho}$ from the algorithm)

$$d_k = \frac{\rho + \lambda}{2\rho}g_k^+ + \frac{\rho - \lambda}{2\rho}g_k^- = H(w_{t,k} - c) + \lambda H\hat{v}_k. \tag{89}$$

Projecting onto the eigenvector $q$ (write $\mu$ for the eigenvalue) yields the scalar exact update for the inner step:

$$x_{t,k+1} = (1 - \eta\mu)\,x_{t,k} + \eta\mu\,c_{t,k} \;-\; \eta\lambda\mu\,\hat{v}_{k,q}. \tag{90}$$

Thus the inner-step perturbation (momentum accumulation error) is exactly

$$\varepsilon_{t,k} \triangleq -\eta\lambda\mu\,\hat{v}_{k,q}. \tag{91}$$

Unrolling the inner loop (as in the standard linear system expansion) gives the closed form for the $k$-step output $y_k \triangleq x_{t,k}$:

$$y_k = a^k x_t + \eta\mu\sum_{r=0}^{k-1}a^{k-1-r}c_{t,r} \;+\; \sum_{r=0}^{k-1}a^{k-1-r}\varepsilon_{t,r}, \tag{92}$$

with $a \triangleq 1 - \eta\mu$. The outer Lookahead update is $x_{t+1} = (1 - \alpha)x_t + \alpha y_k$. Group the data-noise part

$$S \triangleq B_{\text{eff}}\sum_{r=0}^{k-1}a^{k-1-r}c_{t,r}, \tag{93}$$

and the momentum accumulation part

$$E_t \triangleq \alpha\sum_{r=0}^{k-1}a^{k-1-r}\varepsilon_{t,r} = -\alpha\eta\lambda\mu\sum_{r=0}^{k-1}a^{k-1-r}\hat{v}_{r,q}. \tag{94}$$

The variance recursion for the linear iteration yields, at stationarity,

$$v_{\text{CMLR}} = \frac{\text{Var}(S) + \text{Var}(E) + 2\,\text{Cov}(S, E)}{1 - A_{\text{eff}}^2}. \tag{95}$$

Identify $\text{Var}(S)$ to obtain $v_{\text{LR}} = \text{Var}(S)/(1 - A_{\text{eff}}^2)$, which proves the decomposition $v_{\text{CMLR}} = v_{\text{LR}} + \Delta$ with $\Delta$ as stated.

It remains to bound $\text{Var}(E)$ and $\text{Cov}(S, E)$. By linearity and independence assumptions on the sampled $c_{t,r}$ (standard in this noise-model analysis),

$$\text{Var}(E) = \alpha^2 \eta^2 \lambda^2 \mu^2 \text{ Var}\Big( \sum_{r=0}^{k-1} a^{k-1-r} \hat{v}_{r,q} \Big).$$

$$\leq \alpha^2 \eta^2 \lambda^2 \mu^2 \frac{1 - a^{2k}}{1 - a^2} \sup_r \text{Var}(\hat{v}_{r,q}). \tag{96}$$

Then we bound $\text{Var}(\hat{v}_{r,q})$:

$$n(u + s_r + \xi_r^+) = n(u) + Dn_u(s_r + \xi_r^+) + \frac{1}{2} D^2 n_{u+\theta_r^+(s_r+\xi_r^+)}(s_r + \xi_r^+, s_r + \xi_r^+) \tag{97}$$

$$n(u - s_r + \xi_r^-) = n(u) + Dn_u(-s_r + \xi_r^-) + \frac{1}{2} D^2 n_{u+\theta_r^-(-s_r+\xi_r^-)}(-s_r + \xi_r^-, -s_r + \xi_r^-) \tag{98}$$

$$n(g_r^+) + n(g_r^-) = 2n(u) + Dn_u(\xi_r^+ + \xi_r^-) + \underbrace{\frac{1}{2} D^2 n_{u+\theta_r^+(s_r+\xi_r^+)}(s_r + \xi_r^+, s_r + \xi_r^+)}_{R_r^{(1)}}$$

$$+ \underbrace{\frac{1}{2} D^2 n_{u+\theta_r^-(-s_r+\xi_r^-)}(-s_r + \xi_r^-, -s_r + \xi_r^-)}_{R_r^{(2)}} \tag{99}$$

where (By Lemma E.1)

$$\|R_r^{(1)}\| \leq \frac{1}{2} \cdot \frac{3}{\|u + \theta_r^+(s_r + \xi_r^+)\|^2} \|s_r + \xi_r^+\|^2$$

$$\leq \frac{3}{2(\|u\|/2)^2} \|s_r + \xi_r^+\|^2 = 6 \frac{\|s_r + \xi_r^+\|^2}{\|u\|^2} \tag{100}$$

$$\|R_r^{(2)}\| \leq 6 \frac{\| - s_r + \xi_r^-\|^2}{\|u\|^2} \tag{101}$$

$$\|R_r^{(1)}\| + \|R_r^{(2)}\| \leq 6 \frac{2\|s_r\|^2 + 2\|\xi_r^+\|^2 + 2\|s_r\|^2 + 2\|\xi_r^-\|^2}{\|u\|^2}$$

$$= \frac{24\|s_r\|^2 + 12(\|\xi_r^+\|^2 + \|\xi_r^-\|^2)}{\|u\|^2} \tag{102}$$

By substituting into the momentum recursive equation, we obtain:

$$v_{r+1} = \beta v_r + (1 - \beta)n(u) + \frac{1 - \beta}{2} Dn_u(\xi_r^+ + \xi_r^-) + \frac{1 - \beta}{2}(R_r^{(1)} + R_r^{(2)})$$

$$= \beta v_r + (1 - \beta)n(u) + \eta_r + y_{r+1} \tag{103}$$

with $\eta_r \triangleq \frac{1-\beta}{2} Dn_u(\xi_r^+ + \xi_r^-)$, $\rho_r \triangleq \frac{1-\beta}{2}(R_r^{(1)} + R_r^{(2)})$. Thus we have:

$$y_{r+1} = \beta y_r + \eta_r + \rho_r. \tag{104}$$

By $\text{Var}(y_{r+1,q}) = \beta^2 \text{Var}(y_{r,q}) + \text{Var}(\eta_q + \rho_q) + 2\beta\text{Cov}(y_q, \eta_q + \rho_q)$ and $2\beta|\text{Cov}| \leq 2\beta\sqrt{\text{Var}(y)\text{Var}(\eta + \rho)}$, we obtain:

$$\text{Var}(y_q) \leq \frac{\text{Var}(\eta_q) + \sup_r \text{Var}(\rho_{r,q}) + 2\sup_r \sqrt{\text{Var}(\eta_q)\text{Var}(\rho_{r,q})}}{1 - \beta^2} \tag{105}$$

The first term of RHS in Equation 105:

$$\mathrm{Var}(\eta_q) = \mathbb{E}[(q^\top \eta_r)^2] \leq \mathbb{E}[\|\eta_r\|^2] = \left(\frac{1-\beta}{2}\right)^2 \mathbb{E}[\|Dn_u(\xi_r^+ + \xi_r^-)\|^2]$$

$$\leq \left(\frac{1-\beta}{2}\right)^2 \|Dn_u\|_{\mathrm{op}}^2 \cdot \mathbb{E}[\|\xi_r^+ + \xi_r^-\|^2]$$

$$\leq \left(\frac{1-\beta}{2}\right)^2 \cdot \frac{1}{\|u\|^2} \cdot 4\sigma^2 = (1-\beta)^2 \frac{\sigma^2}{\|u\|^2} \tag{106}$$

and the second term of RHS in Equation 105:

$$\mathbb{E}\|\rho_r\|^2 \leq \left(\frac{1-\beta}{2}\right)^2 \cdot \frac{2(24^2\|s_r\|^4) + 2(12^2\mathbb{E}(\|\xi_r^+\|^2 + \|\xi_r^-\|^2)^2)}{\|u\|^4}$$

$$\leq 288(1-\beta)^2 \frac{\|s_r\|^4}{\|u\|^4} + 72(1-\beta)^2 \frac{\mathbb{E}(\|\xi_r^+\|^2 + \|\xi_r^-\|^2)^2}{\|u\|^4}$$

$$\leq 288(1-\beta)^2 \frac{\|s_r\|^4 + M}{\|u\|^4}$$

$$= C_\rho \frac{\|s_r\|^4 + M}{\|u\|^4} \tag{107}$$

where $C_\rho \triangleq 288(1-\beta)^2$. By Assumption B.4, we obtain

$$\sup_r \mathrm{Var}(\rho_{r,q}) \leq C_\rho \left(\frac{\|s_r\|^4 + M}{\|u\|^4}\right) \tag{108}$$

$$\mathrm{Var}(y_q) \leq \frac{1-\beta}{1+\beta} \cdot \frac{\sigma^2}{\|u\|^2} + \frac{C_\rho}{1-\beta^2} \frac{\|s_r\|^4 + M}{\|u\|^4} + \frac{2\sqrt{C_\rho}}{1-\beta^2} \frac{\sigma}{\|u\|} \sqrt{\frac{\|s_r\|^4 + M}{\|u\|^4}} \tag{109}$$

Then we consider connecting $\hat{v}_{r,q}$ and $y_q$:

$$\hat{v}_{r,q} - q^\top n(u) = \left((I - nn^\top)q\right)^\top y_r + r_{r,q}^{(2)}, \tag{110}$$

$$\mathrm{Var}(\hat{v}_{r,q}) \leq 2\,\mathrm{Var}\left(((I - nn^\top)q)^\top y_r\right) + 2\,\mathrm{Var}(r_{r,q}^{(2)})$$

$$\leq 2\,\|(I - nn^\top)q\|^2 \,\|\mathrm{Cov}(y_r)\|_{\mathrm{op}} + 2\,\mathrm{Var}(r_{r,q}^{(2)})$$

$$\leq 2\,\|\mathrm{Cov}(y_r)\|_{\mathrm{op}} + 2\,\mathrm{Var}(r_{r,q}^{(2)}). \tag{111}$$

where

$$\|\mathrm{Cov}(y_r)\|_{\mathrm{op}} \leq \mathrm{trace}(\mathrm{Cov}(y_r)) = \sum_{i=1}^{d} \mathrm{Var}(y_{r,i})$$

$$\leq d \cdot \sup_i \mathrm{Var}(y_{r,i})$$

$$\leq d \cdot \mathrm{Var}(y_q). \tag{112}$$

Considering $\mathrm{Var}(r_{r,q}^{(2)}) \leq \sup_r \mathrm{Var}(\rho_{r,q})$, we have

$$\mathrm{Var}(\hat{v}_{r,q}) \leq 2d \cdot \mathrm{Var}(y_q) + 2C_\rho \frac{\|s_r\|^4 + M}{\|u\|^4} \tag{113}$$

Considering $\|u\| \geq g_{\min} > 0, \|s_r\| \leq \rho\|H\|$, Let $\tau \triangleq \frac{\mathrm{Var}(E)}{\mathrm{Var}(S)}$, we have

$$v_{\mathrm{CMLR}} \leq v_{\mathrm{LR}}(1 + \sqrt{\tau})^2 + \mathcal{O}\left(\frac{\rho^4\|H\|^4 + M}{g_{\min}^4}\right) \tag{114}$$

where $\tau = 2d\frac{\lambda^2(1-\beta)}{g_{\min}^2(1+\beta)}$.

$\square$

Table 4: Performance comparison of CMLR against multi-step optimizers (k=2,5,10) in CNN models (Test Accuracy %).

| Optimizer | CIFAR-10 (Test Accuracy %) | | | |
|---|---|---|---|---|
| | ResNet-18 | WRN-28-10 | VGG-16-BN | PyramidNet-110 |
| Lookbehind-SAM (K=2,5) | $96.95_{\pm0.13}$ $97.09_{\pm0.13}$ | $97.90_{\pm0.12}$ $98.01_{\pm0.11}$ | $96.10_{\pm0.16}$ $96.25_{\pm0.15}$ | $97.95_{\pm0.23}$ $98.09_{\pm0.22}$ |
| GSAM (K=2,5,10) | $96.92_{\pm0.14}$ $97.12_{\pm0.13}$ $97.34_{\pm0.12}$ | $97.88_{\pm0.15}$ $98.05_{\pm0.14}$ $98.26_{\pm0.13}$ | $96.08_{\pm0.16}$ $96.30_{\pm0.15}$ $96.51_{\pm0.14}$ | $97.93_{\pm0.22}$ $98.12_{\pm0.22}$ $98.35_{\pm0.23}$ |
| FMLR (K=2,5,10) | $96.85_{\pm0.13}$ $97.14_{\pm0.12}$ $97.29_{\pm0.11}$ | $97.82_{\pm0.14}$ $98.07_{\pm0.13}$ $98.21_{\pm0.12}$ | $96.02_{\pm0.16}$ $96.31_{\pm0.15}$ $96.46_{\pm0.15}$ | $97.88_{\pm0.21}$ $98.18_{\pm0.20}$ $98.37_{\pm0.19}$ |
| CMLR (K=2,5,10) | $97.00_{\pm0.13}$ $97.46_{\pm0.12}$ $\mathbf{97.84}_{\pm0.11}$ | $97.96_{\pm0.14}$ $98.30_{\pm0.13}$ $\mathbf{98.63}_{\pm0.13}$ | $96.20_{\pm0.17}$ $96.65_{\pm0.17}$ $\mathbf{97.12}_{\pm0.18}$ | $98.00_{\pm0.20}$ $98.50_{\pm0.19}$ $\mathbf{98.91}_{\pm0.11}$ |

| Optimizer | CIFAR-100 (Test Accuracy %) | | | |
|---|---|---|---|---|
| | ResNet-18 | WRN-28-10 | VGG-16-BN | PyramidNet-110 |
| Lookbehind-SAM (K=2,5) | $80.60_{\pm0.18}$ $80.74_{\pm0.17}$ | $83.90_{\pm0.16}$ $84.03_{\pm0.12}$ | $76.90_{\pm0.19}$ $77.02_{\pm0.13}$ | $85.15_{\pm0.18}$ $85.28_{\pm0.15}$ |
| GSAM (K=2,5,10) | $80.58_{\pm0.17}$ $80.75_{\pm0.16}$ $80.86_{\pm0.16}$ | $83.92_{\pm0.16}$ $84.20_{\pm0.15}$ $84.35_{\pm0.13}$ | $76.93_{\pm0.18}$ $77.15_{\pm0.17}$ $77.33_{\pm0.15}$ | $85.18_{\pm0.17}$ $85.40_{\pm0.16}$ $85.56_{\pm0.13}$ |
| FMLR (K=2,5,10) | $80.62_{\pm0.18}$ $80.78_{\pm0.17}$ $80.90_{\pm0.18}$ | $83.88_{\pm0.17}$ $84.10_{\pm0.16}$ $84.23_{\pm0.12}$ | $76.90_{\pm0.18}$ $77.05_{\pm0.17}$ $77.16_{\pm0.15}$ | $85.12_{\pm0.18}$ $85.30_{\pm0.16}$ $85.44_{\pm0.14}$ |
| CMLR (K=2,5,10) | $81.18_{\pm0.16}$ $81.42_{\pm0.15}$ $\mathbf{81.64}_{\pm0.14}$ | $84.30_{\pm0.16}$ $84.56_{\pm0.14}$ $\mathbf{84.84}_{\pm0.11}$ | $77.35_{\pm0.17}$ $77.50_{\pm0.15}$ $\mathbf{77.65}_{\pm0.13}$ | $85.70_{\pm0.17}$ $85.88_{\pm0.15}$ $\mathbf{86.07}_{\pm0.12}$ |

Table 5: Performance comparison of CMLR against baseline optimizers in CNN and ViT models on Tiny-ImageNet (Test Accuracy %).

| Optimizer | Tiny-ImageNet (Test Accuracy %) | | |
|---|---|---|---|
| | ResNet-18 | ViT-Ti | VGG-16-BN |
| SAM | $64.41_{\pm0.53}$ | $37.81_{\pm0.65}$ | $60.68_{\pm1.19}$ |
| CR-SAM | $64.82_{\pm0.51}$ | $38.57_{\pm0.53}$ | $61.28_{\pm1.16}$ |
| Lookbehind-SAM | $65.32_{\pm0.48}$ | $38.21_{\pm0.60}$ | $61.78_{\pm1.19}$ |
| GSAM | $67.07_{\pm0.26}$ | $39.15_{\pm0.58}$ | $63.12_{\pm1.31}$ |
| FMLR | $66.06_{\pm0.48}$ | $38.46_{\pm0.62}$ | $62.33_{\pm1.14}$ |
| **CMLR** | $\mathbf{68.44}_{\pm0.40}$ | $\mathbf{40.23}_{\pm0.57}$ | $\mathbf{65.84}_{\pm1.15}$ |

## F    DETAILED EXPERIMENTAL RESULTS AND SETTING

We present the main results under the lookahead mechanism (K=2,5,10) in Table 4.
For Tiny-ImageNet, we use AdamW with a learning rate of $1 \times 10^{-3}$, weight decay of $5 \times 10^{-5}$, and fix the perturbation strength to $\rho = 0.05$. For CMLR we again keep K = 10, $\beta_s = 0.9$, $\beta_e = 0.99$ fixed throughout all Tiny-ImageNet runs, and use $\alpha = 0.9$ as the default. The choice of $\lambda$ follows the same small-range search as in the CIFAR experiments. The experimental results are shown in the Table 5
The parameter settings for fine-tuning the pre-trained DistilBERT model are shown in Table 6.

Table 6: Per-task hyperparameter configurations for DistilBERT fine-tuning.

| Task | Batch Size | LR | Epochs | Other Params |
|------|-----------|-----|--------|--------------|
| CoLA | 32 | 2e-5 | 10 | |
| SST-2 | 32 | 2e-5 | 3 | |
| MRPC | 16 | 2e-5 | 5 | $\rho \in \{0.001, 0.005, 0.01, 0.05, 0.1\}$ |
| STS-B | 16 | 2e-5 | 5 | $\lambda \in \{0.001, 0.005, 0.01, 0.05, 0.1\}$ |
| QQP | 32 | 2e-5 | 3 | wd $= 0.01$ |
| MNLI | 32 | 2e-5 | 3 | K=2 |
| QNLI | 32 | 2e-5 | 3 | |
| RTE | 16 | 1e-5 | 10 | |

## G  ABLATION STUDY

### G.1  REGULARIZATION STRENGTH

We analyze the sensitivity of CMLR to the regularization strength hyperparameter $\lambda$. A grid search was performed over $\lambda$ in the range $[0.05, 0.15]$ with a 0.01 step size on both CIFAR-10 and CIFAR-100 datasets.

Figure 4 plots the final test accuracy as a function of $\lambda$. The results show that performance is strong and stable across this range on both datasets, with optimal accuracy consistently achieved when $\lambda$ is approximately 0.1. This indicates that a moderate regularization strength is most effective. We therefore select $\lambda = 0.1$ as the default value for all main experiments.

### G.2  INTERPOLATION COEFFICIENT

A key component of our CMLR algorithm is the momentum accumulation strategy used to effi-ciently determine the perturbation vector for the next inner-loop step, $v_{k+1}$. This strategy calculates a weighted average of the normalized "forward" gradient (from the ascent step, $g_k^+$) and "backward" gradient (from the descent step, $g_k^-$). To achieve this, we modify our base momentum accumulation formula from Equation 10 by introducing an interpolation coefficient $\gamma_{interp}$:

$$\beta_k v_k + (1 - \beta_k) \left( \gamma_{interp} \frac{g_k^+}{\|g_k^+\|} + (1 - \gamma_{interp}) \frac{g_k^-}{\|g_k^-\|} \right) \tag{115}$$

Here, $\gamma_{interp}$ balances the influence of the two directions: a value of 1.0 relies entirely on the ascent gradient, while 0.0 relies solely on the descent gradient. We conducted an ablation study to investigate the impact of this coefficient by training a ResNet-18 model on CIFAR-10 while varying $\gamma_{interp}$ from 0.0 to 1.0. As shown in Figure 5, we found that model performance was strong across a range of values, with optimal test accuracy achieved when $\gamma_{interp}$ was approximately 0.1 or 0.8. This indicates that while a blend of both directions is effective, a slight bias towards either the ascent or descent gradient can be beneficial for guiding the subsequent perturbation.

### G.3  SLOW WEIGHTS STEP SIZE

We analyze the impact of the slow weights step size $\alpha$, which controls the outer-loop Lookahead update . We tested $\alpha$ values from 0.7 to 1.0 with a 0.05 interval on CIFAR-10 and CIFAR-100, using a ResNet-18 model with $K = 10$. Each setting was averaged over five runs.

The error bar plot in Figure 6 shows the final test accuracy as a function of $\alpha$. The results indicate that performance is strong and stable when $\alpha$ is in the $[0.85, 1.0]$ range, with the optimal test accuracy achieved at approximately $\alpha = 0.9$. Based on this finding, we use $\alpha = 0.9$ in our main experiments.

### G.4  EFFICIENCY GAINS

In addition to this primary analysis, we provide a direct measurement of the efficiency gains. We compare CMLR against CLR, a variant that computes all three gradients per inner-loop step without momentum. The results, shown in Table 7, confirm that CMLR increases throughput by approxi-mately 45% while matching the final test accuracy of the naive CLR with a negligible difference.

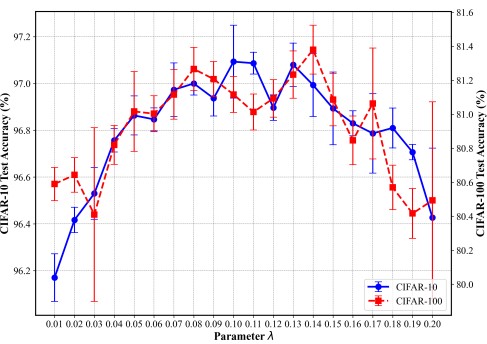

Figure 4: The impact of the hyperparameter $\lambda$ on final test accuracy. The experiment was conducted on ResNet-18 with the CIFAR-10/100 datasets.

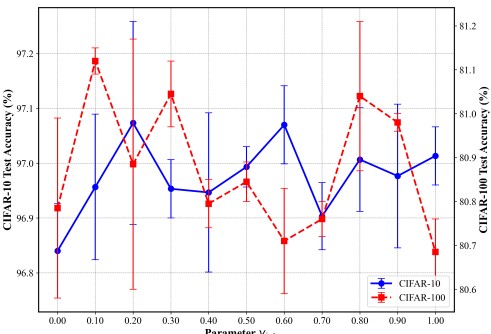

Figure 5: The impact of the hyperparameter $\gamma_{interp}$ on final test accuracy conducted on ResNet-18 with the CIFAR-10 and CIFAR-100 datasets.

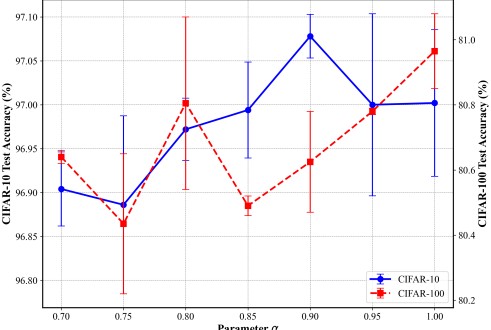

Figure 6: The impact of the hyperparameter $\alpha$ on final test accuracy conducted on ResNet-18 with the CIFAR-10 and CIFAR-100 datasets.

Table 7: Performance and relative training time on CIFAR-10/100.

| Method | CIFAR-10 | | CIFAR-100 | |
|---|---|---|---|---|
| | Acc. (%) | Rel. Train Time | Acc. (%) | Rel. Train Time |
| CLR | 97.17 | 1.47 | 81.15 | 1.43 |
| CMLR | 97.10 | 1.00 | 81.05 | 1.00 |

Table 8: Wall-clock breakdown per parameter update in ResNet34 model, CiFar10 dataset. Multi-step methods are shown with sub-rows for different $K \in \{2, 5, 10\}$. "Forward/Backward" counts the number of total gradient evaluations; each requires one forward and one backward pass. "Parallel" indicates whether gradient evaluations can be done in parallel (e.g., for central difference).

| Optimizer | Setting(K) | Forward/Backward | Peak memory | GPU time | Parallel |
|---|---|---|---|---|---|
| Vanilla (SGD/AdamW) | – | 1 | 1.00× | 1.00× | ✗ |
| SAM (two-pass) | 1 | 2 | 1.28× | 2.18× | ✗ |
| FR (GR view of SAM) | 1 | 2 | 1.25× | 2.26× | ✗ |
| CR (GR view of SAM) | 1 | 2 | 1.25× | 2.14× | ✓ |
| CR-SAM (Curvature Regularized) | 1 | 3 | 1.08× | 2.00× | ✓ |
| Lookbehind-SAM (multi-step) | 2 | 4 | 1.06× | 3.03× | ✗ |
| | 5 | 10 | 1.06× | 6.19× | ✗ |
| | 10 | 20 | 1.06× | 11.48× | ✗ |
| GSAM (multi-step) | 2 | 4 | 1.03× | 3.07× | ✗ |
| | 5 | 10 | 1.03× | 6.25× | ✗ |
| | 10 | 20 | 1.03× | 11.39× | ✗ |
| FMLR (FR-based Lookahead) | 2 | 4 | 1.25× | 3.56× | ✗ |
| | 5 | 10 | 1.25× | 7.48× | ✗ |
| | 10 | 20 | 1.25× | 14.07× | ✗ |
| CMLR (CR-based Lookahead) | 2 | 5 | 1.25× | 3.42× | ✓ |
| | 5 | 11 | 1.25× | 7.19× | ✓ |
| | 10 | 21 | 1.25× | 13.50× | ✓ |

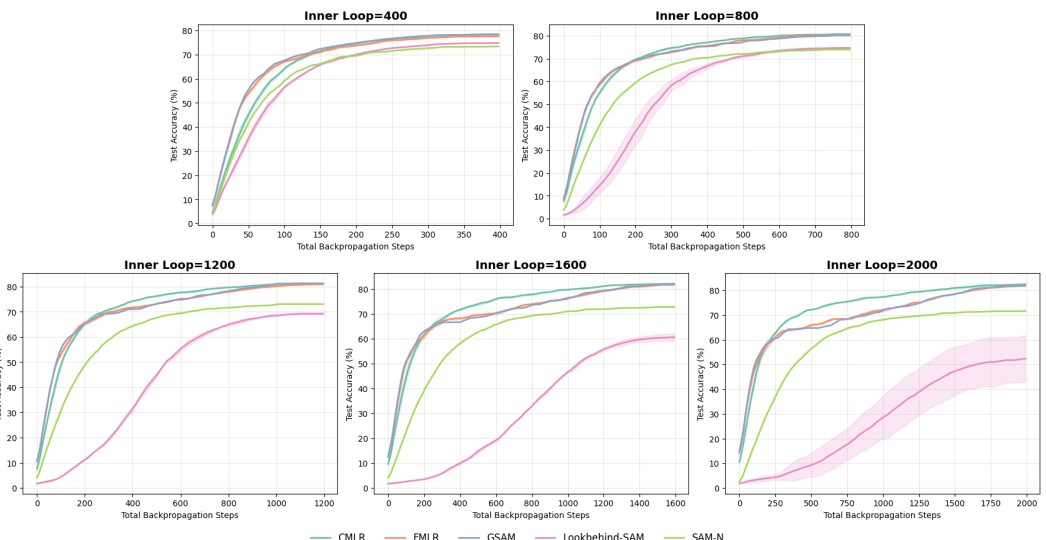

Figure 7: Test accuracy versus backpropagation steps on ResNet-50/CIFAR-100. The comparison involves five algorithms under the same backpropagation budgets (400, 800, 1200, 1600, 2000), illustrating their performance trends.

## DISCLOSURE OF LLM USAGE

We used a large language model (DouBao) solely for minor language polishing. All technical content, methodology, experiments, and analyses were developed entirely by the authors.

