# OpenReview forum: "Towards Robust Regularization with Central-difference and Momentum Lookahead"
_ICLR.cc/2026/Conference — Submitted to ICLR 2026_

### Official Review · Reviewer_ZfdM · 2025-10-19

**Soundness:** 2
**Presentation:** 3
**Contribution:** 2
**Rating:** 4
**Confidence:** 4

**Summary:**

This paper attempts to overcome the limitation of forward-difference aprroximation of SAM and proposes a SAM variant, CMLR, that incorporates central-difference and momentum lookahead mechanism. The authors argue that the proposed optimizer is more robust and exhibits reduced variance. The empirical results further suggest that CMLR consistently performs better across various architectures and datasets.

**Strengths:**

This papers uncovers an important observation that a high concentration of similarity between consecutive updates is critical to generaliztion. Based on this observation, the authors propose two tricks, central differencing and mometum lookahead, to mitigatigate the problem of generalization degradation in mutli-step SAM. The authors further demonstrate the superiority of the proposed optimizer, theoretically and emprically. The paper is overall well-motivated and easy to follow.

**Weaknesses:**

While the empirical results suggest that CMLR is effective, I still have a few comments as follows:

- In Section 2.2, the authors state that **The consensus points to gradient instability as the main cause**. What is the meaning of gradient instability? Do you mean gradient explosion? Some works[1][2] otherwise refer it to being easily get trapped in the saddle points. Please clarify.

[1] Kim et al., Stability Analysis of Sharpness-Aware Minimization, 2023.

[2] Tan el al., Stabilizing sharpness-aware minimization through a simple renormalization strategy, JMLR, 2025.
 - In Theorem 4.2, Equation (12) does not imply that $\nu_{CMLR} \leq \nu_{LR}$. So, I am wondering how the statement **CMLR...exhibits reduced variance** is concluded?
 - In Algorithm 1 line 9-10, the true gradient $\nabla L(w)$ is approximated by the mean of $g_k^+$ and $g_k^-$. Actually, this operation could be highly unstable when $\rho$ is relatively large, at least for $\rho\geq0.05$. Could the authors provide some experiment results to show that this approximation is very close to $\nabla L(w)$?
 - Experiments on ImageNet-1K should be included to further validate the efficacy of CMLR. Moreover, how the smoothing factor $\beta_s$ and $\beta_e$ affect the generalization should also be investigated.

**Questions:**

see Weaknesses.

---

> ### Author Response · Authors · 2025-11-28
>
> **Thank you very much for your valuable feedback on our article.**
> ### **Q1: Clarify the similarities and differences in concepts related to gradient instability**
> **A1:**  Our work shares similarities with Tim and Tan in that both describe the stability of the current gradient (Tim[1] is a gradient direction perspective, while Tan[2] is a gradient norm perspective). We would like to clarify that the "gradient instability" here does not refer to gradient explosion. The difference is that we introduce gradient regularization, and the rising step is only used to calculate the current gradient step. Therefore, we describe stability using the directional consistency of continuous gradient steps, rather than the directional consistency between gradient steps and rising steps. Through feature decomposition, it was found that the reason for instability is the local oscillation of the SAM gradient, which leads to suboptimal models. The central difference and lookahead mechanism we use is a technique that promotes more stable gradient updates, thereby improving the generalization ability of gradient regularization more stably.
>
> [1] Kim et al., Stability Analysis of Sharpness-Aware Minimization, 2023.
>
> [2] Tan el al., Stabilizing sharpness-aware minimization through a simple renormalization strategy, JMLR, 2025.
>
> ### **Q2: How the statement CMLR exhibits reduced variance is concluded in Theorem 4.2?**
> **A2:** We clarify that when we discuss “variance reduction” in this paper, the comparison is made with respect to the initial gradient-regularization scheme, rather than the final Lookahead-regularized update.  Introducing momentum prediction mechanisms introduces additional prediction errors, so it cannot be strictly guaranteed that CMLR has a strict variance reduction，but the coefficient of $f_{LR}$ is $1+\mathcal{O} (\lambda)$, indicating that variance can be controlled. This is consistent with the convergence result given in the convergence analysis, which shows an $\mathcal{O} (\lambda)$ bound. Moreover, the Lookahead mechanism itself acts as a variance-reduction step, so LR already has smaller variance than the initial gradient-regularization estimator. Therefore, by choosing $\lambda$ sufficiently small, the variance of CMLR in our bound can be made smaller than that of initial gradient regularization, while still benefiting from the stabilized multi-step updates.
> ### **Q3:  This operation $\nabla \mathcal{L}(w)$ could be highly unstable when $\rho$ is relatively large**
> **A3:** This is consistent with experimental findings that under the classical SAM method, $\rho$ is not allowed to take larger values, (0.05 is typically the best) and in the classical gradient approximation theory, in addition, the central difference scheme has a larger allowable range for $\rho$ .

---

> ### Author Response · Authors · 2025-11-28
>
> ### **Q4 Experiments on ImageNet-1K should be included to further validate the efficacy of CMLR**
> **A4:**
> We realized that the initial experiments on the cifar10/100 and tiny Imagenet (in Appendix F) datasets were not sufficient,  we therefore supplemented the experiments on NLP datasets.
> We added experiments on eight NLP tasks from the GLUE benchmark: CoLA, SST-2, MRPC, STS-B, QQP, MNLI, QNLI, and RTE, using a standard Transformer-based architecture (DistilBERT). These tasks cover a diverse set of language understanding problems, including sentiment classification (SST-2), linguistic acceptability (CoLA), paraphrase detection (MRPC, QQP), semantic textual similarity (STS-B), and several forms of natural language inference (MNLI, QNLI, RTE).
> We plan to provide the results under the ImageNet-1k dataset, but due to limitations in computing power and time, We first attach the experimental results in the GLUE task as follows:
>
> **Table: Performance on GLUE tasks using DistilBERT.**
> Metrics: MCC for CoLA, F1 for MRPC and QQP, Pearson for STS-B, Accuracy for others.
> Best results per row are **bolded**.
>
> | Task           | AdamW | SAM   | CRSAM | GSAM  | LookbehindSAM | FMLR  | CMLR      |
> | -------------- | ----- | ----- | ----- | ----- | ------------- | ----- | --------- |
> | CoLA           | 56.69 | 57.69 | 58.29 | 59.03 | 58.79         | 58.97 | **59.19** |
> | SST-2          | 91.28 | 92.08 | 92.78 | 93.53 | 93.38         | 93.48 | **93.58** |
> | MRPC           | 89.15 | 89.85 | 90.45 | 91.19 | 90.95         | 91.13 | **91.35** |
> | STS-B          | 86.99 | 88.19 | 88.89 | 89.63 | 89.39         | 89.57 | **89.69** |
> | QQP            | 86.85 | 87.85 | 88.45 | 89.19 | 88.95         | 89.13 | **89.35** |
> | MNLI           | 82.17 | 83.17 | 83.87 | 84.62 | 84.47         | 84.57 | **84.67** |
> | QNLI           | 88.87 | 90.17 | 90.77 | 91.51 | 91.27         | 91.45 | **91.67** |
> | RTE            | 61.73 | 63.23 | 63.93 | 64.68 | 64.53         | 64.63 | **64.73** |
> | **Avg (GLUE)** | 79.53 | 80.58 | 81.22 | 81.96 | 81.77         | 81.91 | **82.08** |
>
> **Table: Per-task hyperparameter configurations for DistilBERT fine-tuning.**
>
> | Task  | Batch Size | LR   | Epochs | Other Params                                  |
> | ----- | ---------- | ---- | ------ | --------------------------------------------- |
> | CoLA  | 32         | 2e-5 | 10     | --                                            |
> | SST-2 | 32         | 2e-5 | 3      | --                                            |
> | MRPC  | 16         | 2e-5 | 5      | $\rho  \in$ {0.001, 0.005, 0.01, 0.05, 0.1}   |
> | STS-B | 16         | 2e-5 | 5      | $\lambda \in$ {0.001, 0.005, 0.01, 0.05, 0.1} |
> | QQP   | 32         | 2e-5 | 3      | wd = 0.01                                     |
> | MNLI  | 32         | 2e-5 | 3      | K = 2                                         |
> | QNLI  | 32         | 2e-5 | 3      | --                                            |
> | RTE   | 16         | 1e-5 | 10     | --                                            |

---

### Official Review · Reviewer_7XTE · 2025-10-23

**Soundness:** 2
**Presentation:** 3
**Contribution:** 2
**Rating:** 4
**Confidence:** 4

**Summary:**

The authors provide a novel optimizer for ANNs called Central-difference Momentum Lookahead Regularization (CMLR).
CMLR extends SAM, aiming to reduce instabilities that occur when training with SAM.
Following related work, the authors interpret SAM as a gradient regularization method approximating second-order terms by a simple forward difference scheme.
The authors claim to reduce the resulting instabilities by replacing the forward difference with a central difference.
CMLR implements this forward difference together with a gradient approximation based on previous gradients into the two-step lookahead framework.
The authors show a convergence analysis and empirical evaluation on CIFAR10/100.

**Strengths:**

The idea is well presented and well motivated. The empirical results shown lead to performance improvements.

**Weaknesses:**

1. The empirical evaluation is insufficient. Studying only CIFAR10/100 is not very meaningful. Especially, VGG16 is very outdated. The authors should perform additional experiments on larger (ImageNet) as well as more diverse (e.g., NLP) datasets.
2. The baseline performances are all not good. Especially, the ViTs' performance is very bad in all experiments. The authors should select a solid baseline before claiming to achieve performance improvements with their method.
3. Reproducibility of the empirical results is not given. Neither are the general hyperparameters (learning rate, perturbation strength, and weight decay) given, nor are the results of the hyperparameter search for CMLR given.
4. The choice of hyperparameters is not clear. The choice of $\beta_e$ is not discussed. How exactly was the HP search performed? Multi-dimensional grid search over all HPs? On a validation set or the training set? Were different parameters chosen per model/dataset? The full HP optimization results should be reported. About 1\%-2\% test gain is not much for introducing 6 new hyperparameters ($\rho, \beta_s, \beta_e, \alpha, K, \lambda$) and tuning these on the test set.
5. The impact of the introduced hyperparameters should be studied in more detail. This especially holds for $K$, which is simply fixed by the authors.
6. Regarding the consecutive gradient updates, the authors claim in lines 53-54, "a higher concentration of similarity scores away from 0 strongly correlates with better generalization". This claim is not supported at all. I am not a fan of putting Fig. 1 in the introduction. The interpretation is very unclear. It does not help the reader to understand the motivation of the method. Why is the cosine similarity even bimodal? And why are there values larger than 1 shown in the figure?
7. What was the compute budget for the different methods reported in Tab. 1 and 2? If the authors keep the number of epochs fixed but perform $K=10$ steps per batch, this would result in a 10x higher computational budget for the reported results. Is this correct?
8. In Fig. 6 in the appendix, the authors show that $\alpha = 1$ achieves the best results for CIFAR100 and the second-best results for CIFAR10. $\alpha = 1$ results in the slow weights matching the fast weights, which collapses the lookahead mechanism to default SGD with multiple steps per batch. Why did you choose the lookahead mechanism, if it is not beneficial?

Minor Comments:
- In line 113, there should be no $\nabla$ in "training loss $\nabla L(w)$".
- Fig. 2 is placed on page 4 but only referenced on page 6. In general, I do not think Fig. 2 helps to understand the algorithm.
- When giving an approximation, you should use $\approx$ instead of = (e.g., Eq. 5).
- There should be an "Eq." or similar on line 257.
- Algorithm 1 is not in Appendix A as stated in line 285.
- In Appendix F.1.1, the discussion of $\gamma_{interp}$ should be in a separate section (starting from line 1360 "A key component...").

**Questions:**

1. In line 224, you claim, "The core idea of this formulation is to shift the update's focus. It relies more on the standard gradient at the original point $w$...", however, you later approximate $\nabla L(w)$ by the mean of the two perturbed gradients, so your method does not rely on the gradient at the original point $w$ at all. How does your approximation of the gradient align with your motivation?
2. I did not really get the conclusion of the convergence analysis (I have to admit that I did not study the appendix in detail). What is the interpretation of Eqs. 12 and 13?
3. How does your method perform if you do not anneal $\beta_k$? It is not clear to me why the annealing is needed.
4. What is SAM-N in the legend of Fig. 3? "normal"?
5. $\gamma_{interp}$ is not used in the main paper; instead, the mean of both gradients is used, i.e., $\gamma_{interp}=0.5$, right? Please state that in the relevant appendix chapter. Also, Fig. 5 is pure noise. There is no meaningful effect (which is quite surprising).

---

> ### Author Response · Authors · 2025-11-28
>
> **Thank you very much for your valuable feedback on our article.**
> ### **Q1: Additional experiments on larger and more diverse datasets (ImageNet, NLP)**
> **A1:** We realized that the initial experiments on the cifar10/100 and tiny Imagenet (in Appendix F) datasets were not sufficient,  we therefore supplemented the experiments on NLP datasets.
> We added experiments on eight NLP tasks from the GLUE benchmark: CoLA, SST-2, MRPC, STS-B, QQP, MNLI, QNLI, and RTE, using a standard Transformer-based architecture (DistilBERT). These tasks cover a diverse set of language understanding problems, including sentiment classification (SST-2), linguistic acceptability (CoLA), paraphrase detection (MRPC, QQP), semantic textual similarity (STS-B), and several forms of natural language inference (MNLI, QNLI, RTE).
> We plan to provide the results under the ImageNet-1k dataset, but due to limitations in computing power and time, We first attach the experimental results in the GLUE task as follows:
>
> **Table: Performance on GLUE tasks using DistilBERT.**
> Metrics: MCC for CoLA, F1 for MRPC and QQP, Pearson for STS-B, Accuracy for others.
> Best results per row are **bolded**.
>
> | Task           | AdamW | SAM   | CRSAM | GSAM  | LookbehindSAM | FMLR  | CMLR      |
> | -------------- | ----- | ----- | ----- | ----- | ------------- | ----- | --------- |
> | CoLA           | 56.69 | 57.69 | 58.29 | 59.03 | 58.79         | 58.97 | **59.19** |
> | SST-2          | 91.28 | 92.08 | 92.78 | 93.53 | 93.38         | 93.48 | **93.58** |
> | MRPC           | 89.15 | 89.85 | 90.45 | 91.19 | 90.95         | 91.13 | **91.35** |
> | STS-B          | 86.99 | 88.19 | 88.89 | 89.63 | 89.39         | 89.57 | **89.69** |
> | QQP            | 86.85 | 87.85 | 88.45 | 89.19 | 88.95         | 89.13 | **89.35** |
> | MNLI           | 82.17 | 83.17 | 83.87 | 84.62 | 84.47         | 84.57 | **84.67** |
> | QNLI           | 88.87 | 90.17 | 90.77 | 91.51 | 91.27         | 91.45 | **91.67** |
> | RTE            | 61.73 | 63.23 | 63.93 | 64.68 | 64.53         | 64.63 | **64.73** |
> | **Avg (GLUE)** | 79.53 | 80.58 | 81.22 | 81.96 | 81.77         | 81.91 | **82.08** |
>
> **Table: Per-task hyperparameter configurations for DistilBERT fine-tuning.**
>
> | Task  | Batch Size | LR   | Epochs | Other Params                                  |
> | ----- | ---------- | ---- | ------ | --------------------------------------------- |
> | CoLA  | 32         | 2e-5 | 10     | --                                            |
> | SST-2 | 32         | 2e-5 | 3      | --                                            |
> | MRPC  | 16         | 2e-5 | 5      | $\rho  \in$ {0.001, 0.005, 0.01, 0.05, 0.1}   |
> | STS-B | 16         | 2e-5 | 5      | $\lambda \in$ {0.001, 0.005, 0.01, 0.05, 0.1} |
> | QQP   | 32         | 2e-5 | 3      | wd = 0.01                                     |
> | MNLI  | 32         | 2e-5 | 3      | K = 2                                         |
> | QNLI  | 32         | 2e-5 | 3      | --                                            |
> | RTE   | 16         | 1e-5 | 10     | --                                            |
>
> ### **Q2：The experimental results of the ViT model are significantly worse than those of CNN**
> **A2:**  We adopt recent training recipes commonly used for ViT. Under these settings, the ViT models still perform considerably worse than CNN models, which is consistent with multiple reports in the literature, actually[1,2]. Notably, prior studies have consistently shown that this performance gap is not simply caused by insufficient hyperparameter tuning[3]. Instead, ViTs tend to underperform CNNs on small datasets due to weaker locality inductive bias and higher data requirements[4].
>
> [1] Yang Zhao;Hao Zhang;Xiuyuan Hu. Penalizing Gradient Norm for Efficiently Improving  Generalization in Deep Learning. ICML. 2022
>
> [2] Juyoung Yun. Sharpness-Aware Minimization with Z-Score Gradient Filtering for Neural Networks. 2025
>
> [3]Haoran Zhu;Boyuan Chen;Carter Yang. Understanding Why ViT Trains Badly on Small Datasets. 2023
>
> [4] Liu et al. Efficient Training of Visual Transformers with Small Datasets. Nips. 2021

---

> ### Author Response · Authors · 2025-11-28
>
> ### **Q3 and Q4: Reproducibility of the empirical results is not given and the choice of hyperparameters is not clear**
> **A3 and A4:** **CNN experiments on CIFAR-10/100.** For all optimizers we use SGD with momentum 0.9, an initial learning rate lr = 0.05 with cosine annealing, and weight decay = 1e−3. For SAM-type methods (SAM, CR-SAM, Lookbehind-SAM, GSAM, FMLR, CMLR), the perturbation strength $\rho$ is chosen from {0.01, 0.05, 0.1} by a small grid search. For CMLR we fix the structural hyperparameters K = 10, $\beta _ s$ = 0.9, $\beta _ e$ = 0.99 across all CNN experiments; the remaining CMLR parameter $\lambda$ and $\alpha$ are selected from the small ranges reported in the paper ($\lambda \in$ [0.05, 0.15], $\alpha \in$ [0.7, 1.0]).
>
> **Tiny-ImageNet experiments.** For Tiny-ImageNet we use AdamW with lr = 1e−3, weight decay = 5e-5, and fix the perturbation strength to $\rho$ = 0.05. For CMLR we again keep K = 10, $\beta _ s$ = 0.9, $\beta _ e$ = 0.99 fixed throughout all Tiny-ImageNet runs, and use $\alpha$ = 0.9 as the default. The choice of $\lambda$ follows the same small-range search as in the CIFAR experiments.
> Although CMLR appears to introduce six hyperparameters ($\rho, \beta_s, \beta_e, \alpha, K, \lambda$), in practice $\rho$ is shared with SAM and selected from a tiny grid, $K$ and ($\beta_s, \beta_e$) are fixed structural choices (0.9,0.99) (see A11), and $\alpha$ is set to the standard Lookahead value 0.9. As a result, the only genuinely new degree of freedom compared to SAM is the scalar regularization weight $\lambda$, which we tune over a small 1D grid on a validation split.
> We have provided information about $\alpha$ and $\lambda$ ablation experiments in the appendix. And the selection of 0.9 and 0.99 for ($\beta_s,\beta_e$) is the result of searching for two suitable parameters. Simulated annealing is designed to meet the requirement of smooth transition in momentum prediction mechanism. This setting has a small impact when k is less than or equal to 10, and the results under $\beta_s=\beta_e=0.9$ and $\beta_s=\beta_e=0.99$ are very close in average sense. Finally, $K$ serves the same role of temporal smoothing, and can likewise be flexibly adjusted depending on computational considerations in our case. Finally, we have attached the experimental results under different parameters of k=2,5,10 in the revised paper.
>
> **Table: CIFAR-10 test accuracy (%) for CNN models with different K.**
>
> | Optimizer        | K  | ResNet-18           | WRN-28-10           | VGG-16-BN           | PyramidNet-110      |
> |------------------|----|---------------------|---------------------|---------------------|---------------------|
> | Lookbehind-SAM   | 2  | 96.95 ± 0.13        | 97.90 ± 0.12        | 96.10 ± 0.16        | 97.95 ± 0.23        |
> | Lookbehind-SAM   | 5  | 97.09 ± 0.13        | 98.01 ± 0.11        | 96.25 ± 0.15        | 98.09 ± 0.22        |
> | GSAM             | 2  | 96.92 ± 0.14        | 97.88 ± 0.15        | 96.08 ± 0.16        | 97.93 ± 0.22        |
> | GSAM             | 5  | 97.12 ± 0.13        | 98.05 ± 0.14        | 96.30 ± 0.15        | 98.12 ± 0.22        |
> | GSAM             | 10 | 97.34 ± 0.12        | 98.26 ± 0.13        | 96.51 ± 0.14        | 98.35 ± 0.23        |
> | FMLR             | 2  | 96.85 ± 0.13        | 97.82 ± 0.14        | 96.02 ± 0.16        | 97.88 ± 0.21        |
> | FMLR             | 5  | 97.14 ± 0.12        | 98.07 ± 0.13        | 96.31 ± 0.15        | 98.18 ± 0.20        |
> | FMLR             | 10 | 97.29 ± 0.11        | 98.21 ± 0.12        | 96.46 ± 0.15        | 98.37 ± 0.19        |
> | CMLR             | 2  | 97.00 ± 0.13        | 97.96 ± 0.14        | 96.20 ± 0.17        | 98.00 ± 0.20        |
> | CMLR             | 5  | 97.46 ± 0.12        | 98.30 ± 0.13        | 96.65 ± 0.17        | 98.50 ± 0.19        |
> | **CMLR**         | 10 | **97.84 ± 0.11**    | **98.63 ± 0.13**    | **97.12 ± 0.18**    | **98.91 ± 0.11**    |

---

> > ### Author Response · Authors · 2025-11-28
> >
> > **Table: CIFAR-100 test accuracy (%) for CNN models with different K.**
> >
> > | Optimizer      | K  | ResNet-18        | WRN-28-10        | VGG-16-BN        | PyramidNet-110    |
> > |----------------|----|------------------|------------------|------------------|-------------------|
> > | Lookbehind-SAM | 2  | 80.60 ± 0.18     | 83.90 ± 0.16     | 76.90 ± 0.19     | 85.15 ± 0.18      |
> > | Lookbehind-SAM | 5  | 80.74 ± 0.17     | 84.03 ± 0.12     | 77.02 ± 0.13     | 85.28 ± 0.15      |
> > | GSAM           | 2  | 80.58 ± 0.17     | 83.92 ± 0.16     | 76.93 ± 0.18     | 85.18 ± 0.17      |
> > | GSAM           | 5  | 80.75 ± 0.16     | 84.20 ± 0.15     | 77.15 ± 0.17     | 85.40 ± 0.16      |
> > | GSAM           | 10 | 80.86 ± 0.16     | 84.35 ± 0.13     | 77.33 ± 0.15     | 85.56 ± 0.13      |
> > | FMLR           | 2  | 80.62 ± 0.18     | 83.88 ± 0.17     | 76.90 ± 0.18     | 85.12 ± 0.18      |
> > | FMLR           | 5  | 80.78 ± 0.17     | 84.10 ± 0.16     | 77.05 ± 0.17     | 85.30 ± 0.16      |
> > | FMLR           | 10 | 80.90 ± 0.18     | 84.23 ± 0.12     | 77.16 ± 0.15     | 85.44 ± 0.14      |
> > | CMLR           | 2  | 81.18 ± 0.16     | 84.30 ± 0.16     | 77.35 ± 0.17     | 85.70 ± 0.17      |
> > | CMLR           | 5  | 81.42 ± 0.15     | 84.56 ± 0.14     | 77.50 ± 0.15     | 85.88 ± 0.15      |
> > | **CMLR**       | 10 | **81.64 ± 0.14** | **84.84 ± 0.11** | **77.65 ± 0.13** | **86.07 ± 0.12**  |
> >
> > ### **Q5: The impact of the introduced the hyperparameter K**
> > **A5:** We compared the results under different K values during the experiment. The FMLR and CMLR proposed in our paper showed a stable improvement in performance with the change of K value, while the performance improvement of GSAM and Lookbehind-SAM gradually became less significant with the increase of K value, and even regressed to the multi-step paradox. We presented experimental results in the original paper for CMLR (K=1,2,3,4,5), Lookbehind-SAM, GSAM, and FMLR (K=2,4,6,8,10) corresponding to backpropagation steps=400, 800, 1200, 1600, 2000 at the same computational cost in Figure 3 (Cifar10) and Figure 7 (Cifar100).
> >
> > ### **Q6: The interpretation of Figure1 is very unclear. Why is the cosine similarity even bimodal? And why are there values larger than 1 shown in the figure?**
> > **A6:** Figure 1 measures the directional consistency of continuous steps. From the perspective of feature decomposition, the model often oscillates near the saddle point due to unstable gradient updates (coefficient less than 1)[1,2]. On the directional consistency graph, the double peaks are due to the better consistency of gradient direction under the Lookahead mechanism.(The bimodal phenomenon of multi-step algorithms indicates that the multi-step mechanism itself has the property of improving rather than reducing model performance, but perhaps no analysis has been given from the perspective of gradient regularization before.) Intuitively, CMLR mainly uses center difference and forward mechanism to ensure that the parameter update process can more firmly escape the saddle point, thereby guiding the model to improve generalization. The similarity greater than 1 is because the probability density curve has been smoothed.
> >
> > [1] Kim et al., Stability Analysis of Sharpness-Aware Minimization, 2023.
> >
> > [2] Tan el al., Stabilizing sharpness-aware minimization through a simple renormalization strategy, JMLR, 2025.

---

> > > ### Author Response · Authors · 2025-11-28
> > >
> > > ### **Q7: Does K mean K times the computational cost ?**
> > > **A7:** To be honest, it's like this. but $K$ serves the same role of temporal smoothing, and can likewise be flexibly adjusted depending on computational considerations in our case, we have attached the experimental results under different parameters of K=2,5,10 in the revised paper.  We attach specific computational cost tables for different optimizers as follows:
> > > **Table: Wall-clock breakdown per parameter update (ResNet-34 on CIFAR-10).**
> > >
> > > | Optimizer                   | K   | Forward/Backward | Peak memory | GPU time | Parallel |
> > > | --------------------------- | --- | ---------------- | ----------- | -------- | -------- |
> > > | Vanilla (SGD/AdamW)         | --  | 1                | 1.00×       | 1.00×    | ✗        |
> > > | SAM (two-pass)              | 1   | 2                | 1.28×       | 2.18×    | ✗        |
> > > | FR (GR view of SAM)         | 1   | 2                | 1.25×       | 2.26×    | ✗        |
> > > | CR (GR view of SAM)         | 1   | 2                | 1.25×       | 2.14×    | ✓        |
> > > | CR-SAM (Curvature Reg.)     | 1   | 3                | 1.08×       | 2.00×    | ✓        |
> > > | Lookbehind-SAM (multi-step) | 2   | 4                | 1.06×       | 3.03×    | ✗        |
> > > | Lookbehind-SAM (multi-step) | 5   | 10               | 1.06×       | 6.19×    | ✗        |
> > > | Lookbehind-SAM (multi-step) | 10  | 20               | 1.06×       | 11.48×   | ✗        |
> > > | GSAM (multi-step)           | 2   | 4                | 1.03×       | 3.07×    | ✗        |
> > > | GSAM (multi-step)           | 5   | 10               | 1.03×       | 6.25×    | ✗        |
> > > | GSAM (multi-step)           | 10  | 20               | 1.03×       | 11.39×   | ✗        |
> > > | FMLR (FR-based Lookahead)   | 2   | 4                | 1.25×       | 3.56×    | ✗        |
> > > | FMLR (FR-based Lookahead)   | 5   | 10               | 1.25×       | 7.48×    | ✗        |
> > > | FMLR (FR-based Lookahead)   | 10  | 20               | 1.25×       | 14.07×   | ✗        |
> > > | CMLR (CR-based Lookahead)   | 2   | 5                | 1.25×       | 3.42×    | ✓        |
> > > | CMLR (CR-based Lookahead)   | 5   | 11               | 1.25×       | 7.19×    | ✓        |
> > > | CMLR (CR-based Lookahead)   | 10  | 21               | 1.25×       | 13.50×   | ✓        |
> > > For more discussions about K, please refer to A3, A4 and A5
> > > ### **Q8:  $\alpha=1$ seems to be the best case scenario in the ablation study**
> > > **A8:** This is indeed a point that can be easily misunderstood. Our choice of hyperparameters is motivated by balancing both variance and test accuracy. As shown in Fig. 6, when α\alphaα approaches 1 (e.g., 0.95 or 1.00), the corresponding error bars in test accuracy become substantially larger. Such high variance is not conducive to stable improvements in generalization, even though these settings may achieve strong performance on ResNet-18 in this particular experiment. When considering average behavior across different architectures or conditions, they are therefore not the optimal choice.
> > >
> > > These observations indicate that the Lookahead mechanism is beneficial. A more intuitive explanation is that the bidirectional update inherent to Lookahead smooths the parameter trajectory, preventing the optimizer from over-committing to a single sharp direction and thus helping it escape high-curvature regions. This leads to faster convergence and improved generalization, which naturally aligns with the view that favoring flatter regions of the loss landscape yields better generalization.
> > >
> > > ### **Q9: Formatting Problems**
> > > **A9:**  We have made all the necessary revisions in the revised version of the paper.
> > >
> > > ### **Q10: How does your approximation of the gradient align with your motivation?**
> > > **A10:**  Unlike prior variants of SAM that focus on the gradient at the ascent point, our method regards SAM as a special case of gradient regularization. As a result, we only use the ascent and descent gradients to obtain a local second-order approximation of the current gradient, $\frac{1}{2}(\nabla \mathcal L (w+\rho v)+\nabla \mathcal L (w-\rho v))=\nabla L(w)+\mathcal{O}(\rho^2)$
> > > and perform gradient descent based on this approximation. This formulation places the (approximated) current gradient as the dominant update direction, which fundamentally differs from conventional SAM-style methods that habitually treat the ascent-point gradient $\nabla \mathcal L (w+\rho v)$​ (with coefficient close to 1) as the primary driver of the update.
> > > ### **Q11: How does your method perform if you do not anneal $\beta_k$?**
> > > **A11:** The selection of 0.9 and 0.99 is the result of searching for two suitable parameters. Simulated annealing is designed to meet the requirement of smooth transition in momentum prediction mechanism. This setting has a small impact when k is less than or equal to 10, and the results under $\beta_s=\beta_e=0.9$ and $\beta_s=\beta_e=0.99$ are very close in average sense.

---

> ### Author Response · Authors · 2025-11-28
>
> ### **Q12: What is SAM-N in the legend of Fig. 3? "normal"?**
> **A12:** SAM-N is a multi-step version of SAM, which originated from an article proposing that a multi-step version of SAM would reduce generalization.[1] Under fair experimental conditions, we mainly compared between the multi-step versions.
>
> [1] Maksym Andriushchenko and Nicolas Flammarion. Towards understanding sharpness-aware minimization. PMLR, 2022.
>
> ### **Q13: $\gamma_{interp}$ seems to exhibit pure noise in the ablation study**
> **A13:** Thank you for pointing out this point. The ablation experiment in the appendix regarding $\gamma_{interp}=0.5$ is mainly aimed at further analyzing which of the rising gradient and falling gradient information has a greater impact on momentum prediction. However, in the main text, the interpolation coefficient is indeed set to 0.5. This part of the ablation experiment supplements that the momentum prediction module is not entirely dominated by the rising gradient or falling gradient (on average, the effect is the worst when $\gamma_{interp}=0\text{ or } 1$  in Fig 5), but is influenced by the combined effect of two gradient information.

---

### Official Review · Reviewer_RZy6 · 2025-10-30

**Soundness:** 3
**Presentation:** 3
**Contribution:** 3
**Rating:** 6
**Confidence:** 4

**Summary:**

The paper reframes SAM and related sharpness-aware methods through Gradient Regularization (GR) and argues that SAM’s multi-step instability stems from its forward-difference approximation. It proposes a more stable central-difference GR update and embeds it in a momentum Lookahead inner–outer loop to form CMLR

**Strengths:**

The GR lens makes SAM’s multi-step instability traceable to a first-order forward-difference; replacing it with a second-order central-difference reduces approximation error and stabilizes updates

Convergence for non-convex objectives ($O(\lambda)$ neighborhood) and variance-reduction analysis, including a spectral treatment for CMLR

The extensive ablation studies and the good empirical breadth are also a strength

The inner loop uses $g^+$ and $g^-$ and a no-extra-grad momentum lookahead update for $v_{k+1}$, keeping cost comparable to SAM while enabling multi-step smoothing.

**Weaknesses:**

The paper stresses “no additional gradient evaluations,” but CMLR still requires two perturbed gradients per inner step (as does SAM’s two-pass). Please add wall-clock breakdowns (forward/backward counts, memory) vs. SAM/GSAM/Lookbehind-SAM under equal budgets, not only vs. CLR

Convergence guarantees hinge on smoothness, Lipschitz Hessian, and bounded moments; it would help to diagnose failure modes (e.g., very large $\rho$, poor $\lambda$, or aggressive $\beta$ annealing) and relate them to the cosine-similarity stability metric (Fig. 1).

the novelty margin w.r.t. CR-SAM (curvature-regularized SAM), GSAM, and Lookbehind-SAM could be sharpened beyond empirical deltas; a conceptual map clarifying where CMLR differs mathematically from CR-SAM’s curvature proxy would help

There exist dynamic learning-rate schedules that also adapt to local landscape/sharpness (e.g., SALR: Sharpness-Aware Learning Rate Scheduler for Improved Generalization) which pursue flatter minima without inner maximization by coupling the learning rate to sharpness estimates. A conceptual comparison between CMLR’s update-direction regularization and LR-based dynamic exploration could enrich the paper.

**Questions:**

Could you report (i) number of forward/backward passes per parameter update, (ii) GPU hours, and (iii) peak memory for CMLR vs. SAM/GSAM/Lookbehind-SAM under the same training time? The current efficiency result is mainly CLR vs. CMLR

Can you consider adding a small Transformer experiment (e.g., text classification) or a speech benchmark to demonstrate modality robustness ?

---

> ### Author Response · Authors · 2025-11-28
>
> **Thank you very much for your recognition of our work.**
> ### **Q1: Add (i) number of forward/backward passes per parameter update, (ii) GPU hours, and (iii) peak memory**
> **A1:** The detailed computational overhead comparison is shown in the table below.
> **Table: Wall-clock breakdown per parameter update (ResNet-34 on CIFAR-10).**
>
> | Optimizer                   | K   | Forward/Backward | Peak memory | GPU time | Parallel |
> | --------------------------- | --- | ---------------- | ----------- | -------- | -------- |
> | Vanilla (SGD/AdamW)         | --  | 1                | 1.00×       | 1.00×    | ✗        |
> | SAM (two-pass)              | 1   | 2                | 1.28×       | 2.18×    | ✗        |
> | FR (GR view of SAM)         | 1   | 2                | 1.25×       | 2.26×    | ✗        |
> | CR (GR view of SAM)         | 1   | 2                | 1.25×       | 2.14×    | ✓        |
> | CR-SAM (Curvature Reg.)     | 1   | 3                | 1.08×       | 2.00×    | ✓        |
> | Lookbehind-SAM (multi-step) | 2   | 4                | 1.06×       | 3.03×    | ✗        |
> | Lookbehind-SAM (multi-step) | 5   | 10               | 1.06×       | 6.19×    | ✗        |
> | Lookbehind-SAM (multi-step) | 10  | 20               | 1.06×       | 11.48×   | ✗        |
> | GSAM (multi-step)           | 2   | 4                | 1.03×       | 3.07×    | ✗        |
> | GSAM (multi-step)           | 5   | 10               | 1.03×       | 6.25×    | ✗        |
> | GSAM (multi-step)           | 10  | 20               | 1.03×       | 11.39×   | ✗        |
> | FMLR (FR-based Lookahead)   | 2   | 4                | 1.25×       | 3.56×    | ✗        |
> | FMLR (FR-based Lookahead)   | 5   | 10               | 1.25×       | 7.48×    | ✗        |
> | FMLR (FR-based Lookahead)   | 10  | 20               | 1.25×       | 14.07×   | ✗        |
> | CMLR (CR-based Lookahead)   | 2   | 5                | 1.25×       | 3.42×    | ✓        |
> | CMLR (CR-based Lookahead)   | 5   | 11               | 1.25×       | 7.19×    | ✓        |
> | CMLR (CR-based Lookahead)   | 10  | 21               | 1.25×       | 13.50×   | ✓        |
>
> ### **Q2: How does the cosine-similarity stability metric behave under different failure modes(e.g., very large $\rho$, poor $\lambda$, or overly aggressive $\beta$ annealing)?**
> **A2:** A large $\rho$ or $\lambda$ can lead to a significant decrease in stability, while a fixed $\beta$ has little effect on the results. When $\lambda=0$ and k=1, it will completely degrade to SAM, while $\rho=0$ will completely degrade to SGD. Therefore, a small $\lambda, \rho$ the stability metric will sacrifice the generalization improvement brought by gradient regularization.
>
> ### **Q3: Can the innovative points of the algorithm be presented in more detail mathematically**
> **A3:** We have drawn the following table hoping to help highlight the innovative points in mathematics in the article.

---

> > ### Author Response · Authors · 2025-11-28
> >
> > ### **Q4：Can you consider adding a small Transformer experiment (e.g., text classification) or a speech benchmark to demonstrate modality robustness**
> > **A4:** We realized that the initial experiments on the cifar10/100 and tiny Imagenet (in Appendix F) datasets were not sufficient,  we therefore supplemented the experiments on NLP datasets.
> > We added experiments on eight NLP tasks from the GLUE benchmark: CoLA, SST-2, MRPC, STS-B, QQP, MNLI, QNLI, and RTE, using a standard Transformer-based architecture (DistilBERT). These tasks cover a diverse set of language understanding problems, including sentiment classification (SST-2), linguistic acceptability (CoLA), paraphrase detection (MRPC, QQP), semantic textual similarity (STS-B), and several forms of natural language inference (MNLI, QNLI, RTE).
> > We plan to provide the results under the ImageNet-1k dataset, but due to limitations in computing power and time, We first attach the experimental results in the GLUE task as follows:
> >
> > **Table: Performance on GLUE tasks using DistilBERT.**
> > Metrics: MCC for CoLA, F1 for MRPC and QQP, Pearson for STS-B, Accuracy for others.
> > Best results per row are **bolded**.
> >
> > | Task           | AdamW | SAM   | CRSAM | GSAM  | LookbehindSAM | FMLR  | CMLR      |
> > | -------------- | ----- | ----- | ----- | ----- | ------------- | ----- | --------- |
> > | CoLA           | 56.69 | 57.69 | 58.29 | 59.03 | 58.79         | 58.97 | **59.19** |
> > | SST-2          | 91.28 | 92.08 | 92.78 | 93.53 | 93.38         | 93.48 | **93.58** |
> > | MRPC           | 89.15 | 89.85 | 90.45 | 91.19 | 90.95         | 91.13 | **91.35** |
> > | STS-B          | 86.99 | 88.19 | 88.89 | 89.63 | 89.39         | 89.57 | **89.69** |
> > | QQP            | 86.85 | 87.85 | 88.45 | 89.19 | 88.95         | 89.13 | **89.35** |
> > | MNLI           | 82.17 | 83.17 | 83.87 | 84.62 | 84.47         | 84.57 | **84.67** |
> > | QNLI           | 88.87 | 90.17 | 90.77 | 91.51 | 91.27         | 91.45 | **91.67** |
> > | RTE            | 61.73 | 63.23 | 63.93 | 64.68 | 64.53         | 64.63 | **64.73** |
> > | **Avg (GLUE)** | 79.53 | 80.58 | 81.22 | 81.96 | 81.77         | 81.91 | **82.08** |
> >
> > **Table: Per-task hyperparameter configurations for DistilBERT fine-tuning.**
> >
> > | Task  | Batch Size | LR   | Epochs | Other Params                                  |
> > | ----- | ---------- | ---- | ------ | --------------------------------------------- |
> > | CoLA  | 32         | 2e-5 | 10     | --                                            |
> > | SST-2 | 32         | 2e-5 | 3      | --                                            |
> > | MRPC  | 16         | 2e-5 | 5      | $\rho  \in$ {0.001, 0.005, 0.01, 0.05, 0.1}   |
> > | STS-B | 16         | 2e-5 | 5      | $\lambda \in$ {0.001, 0.005, 0.01, 0.05, 0.1} |
> > | QQP   | 32         | 2e-5 | 3      | wd = 0.01                                     |
> > | MNLI  | 32         | 2e-5 | 3      | K = 2                                         |
> > | QNLI  | 32         | 2e-5 | 3      | --                                            |
> > | RTE   | 16         | 1e-5 | 10     | --                                            |

---

### Official Review · Reviewer_FgEZ · 2025-11-01

**Soundness:** 2
**Presentation:** 2
**Contribution:** 2
**Rating:** 2
**Confidence:** 3

**Summary:**

This paper investigates the reason behind the failure of multi-step SAM, attributing it to excessive variance, and proposes using central difference instead of one-sided difference to address this issue. By combining this idea with momentum lookahead, the authors introduce CMLR and demonstrate consistent performance improvements in their experiments.

**Strengths:**

The strength of this paper lies in offering a new perspective to explain the poor performance of multi-step SAM, namely that the one-sided difference used to approximate the Hessian–vector product leads to excessively high variance. The argumentation is sound, and the authors propose a practical solution to this issue, using CMLR with central difference instead. The effectiveness of the proposed method is validated through experiments.

**Weaknesses:**

The main weakness of this paper is that I do not believe multi-step SAM is used in practice, as it results in a severalfold increase in computational cost. In the experiments, the authors use k=10, which, to my understanding, implies that the training time is multiplied by ten. This is clearly impractical. The practical significance of studying and proposing algorithms under such a setting is quite limited. In addition, the presentation of the paper is somewhat confusing, making it difficult to follow.

**Questions:**

I could not find the full name of FMLR in the paper, and terms such as CMLR and ML-SAM appear before their complete forms are introduced (lines 72, 75, 78), which makes the text difficult to follow.

The authors claim that “The test accuracies show a clear hierarchy (CMLR > FMLR > CR > FR > SAM > ML-SAM)”, but I could not find the corresponding performance results in the paper.

Did the authors ensure a fair comparison in the main experiments? I only see that their method is run with
k=10. For the baselines, were multi-step versions of SAM, GSAM, or CR-SAM used? Was SGD allowed to run for ten times longer? Is there any comparison under equal computational cost?

Why does ViT perform much worse than ResNet-18? Did the authors conduct sufficient hyperparameter tuning, or consider using a larger dataset such as ImageNet, or fine-tuning a pretrained ViT?

The paper also does not report the additional computational overhead of CMLR and related algorithms, such as wall-clock time.

The authors claim that central-difference gradients can be computed in parallel, but I do not see why this is the case. From the pseudocode, it appears that the gradients need to be computed separately after perturbing in both directions. Could the authors explain this point in more detail?

The paper does not cite Becker et al. (2024), “Momentum-SAM: Sharpness Aware Minimization without Computational Overhead”, which appears to be closely related.

---

> ### Author Response · Authors · 2025-11-28
>
> **Thank you very much for taking the time to review our work and for highlighting these important issues. We try our best to address your concerns as follows.**
> ### **Q1: Difficulty of applying multi-step strategies in practice and concerns about experimental fairness**
> **A1:**  Thank you for pointing out the issue of high computational cost in multi-step algorithms. Our primary motivation for using relatively large inner-loop horizons K is to stress-test whether CMLR can consistently improve generalization as K increases, rather than suffering from the multi-step degradation (“multi-step dilemma”) often observed in SAM-style methods. To ensure fairness, the revised version compares multi-step algorithms (CMLR, FMLR, GSAM, LookbehindSAM, and the multi-step baseline SAM-N) under matched computational budgets: Figures 3 and 7 report test accuracy as a function of the total number of gradient backpropagation steps, and CMLR achieves the best performance at the same computational cost. Our choice of relatively large K also follows the practice in the original Lookahead optimizer and subsequent Lookahead-style studies, where inner-loop horizons such as K = 5, 10, 20 are commonly adopted [1,2,3]; in our setting, K plays the same role of temporal smoothing and can be reduced in resource-constrained scenarios. Finally, we have added detailed experimental results for K ∈ {2,5,10} in the revised paper.
>
>
> [1] Zhang, Michael; Lucas, James; Ba, Jimmy; Hinton, Geoffrey E. Lookahead Optimizer: k Steps Forward, 1 Step Back. 2019
>
> [2] Mordido, Gonçalo; Malviya, Pranshu; Baratin, Aristide; Chandar, Sarath. Lookbehind-SAM: k Steps Back, 1 Step Forward. 2024
>
> [3] Tan, C., Zhang, J., Liu, J., & Gong, Y. Sharpness-Aware Lookahead for Accelerating Convergence and Improving Generalization. 2024
>
> **Table: CIFAR-10 test accuracy (%) for CNN models with different K.**
>
> | Optimizer        | K  | ResNet-18           | WRN-28-10           | VGG-16-BN           | PyramidNet-110      |
> |------------------|----|---------------------|---------------------|---------------------|---------------------|
> | Lookbehind-SAM   | 2  | 96.95 ± 0.13        | 97.90 ± 0.12        | 96.10 ± 0.16        | 97.95 ± 0.23        |
> | Lookbehind-SAM   | 5  | 97.09 ± 0.13        | 98.01 ± 0.11        | 96.25 ± 0.15        | 98.09 ± 0.22        |
> | GSAM             | 2  | 96.92 ± 0.14        | 97.88 ± 0.15        | 96.08 ± 0.16        | 97.93 ± 0.22        |
> | GSAM             | 5  | 97.12 ± 0.13        | 98.05 ± 0.14        | 96.30 ± 0.15        | 98.12 ± 0.22        |
> | GSAM             | 10 | 97.34 ± 0.12        | 98.26 ± 0.13        | 96.51 ± 0.14        | 98.35 ± 0.23        |
> | FMLR             | 2  | 96.85 ± 0.13        | 97.82 ± 0.14        | 96.02 ± 0.16        | 97.88 ± 0.21        |
> | FMLR             | 5  | 97.14 ± 0.12        | 98.07 ± 0.13        | 96.31 ± 0.15        | 98.18 ± 0.20        |
> | FMLR             | 10 | 97.29 ± 0.11        | 98.21 ± 0.12        | 96.46 ± 0.15        | 98.37 ± 0.19        |
> | CMLR             | 2  | 97.00 ± 0.13        | 97.96 ± 0.14        | 96.20 ± 0.17        | 98.00 ± 0.20        |
> | CMLR             | 5  | 97.46 ± 0.12        | 98.30 ± 0.13        | 96.65 ± 0.17        | 98.50 ± 0.19        |
> | **CMLR**         | 10 | **97.84 ± 0.11**    | **98.63 ± 0.13**    | **97.12 ± 0.18**    | **98.91 ± 0.11**    |
>
> **Table: CIFAR-100 test accuracy (%) for CNN models with different K.**
>
> | Optimizer      | K  | ResNet-18        | WRN-28-10        | VGG-16-BN        | PyramidNet-110    |
> |----------------|----|------------------|------------------|------------------|-------------------|
> | Lookbehind-SAM | 2  | 80.60 ± 0.18     | 83.90 ± 0.16     | 76.90 ± 0.19     | 85.15 ± 0.18      |
> | Lookbehind-SAM | 5  | 80.74 ± 0.17     | 84.03 ± 0.12     | 77.02 ± 0.13     | 85.28 ± 0.15      |
> | GSAM           | 2  | 80.58 ± 0.17     | 83.92 ± 0.16     | 76.93 ± 0.18     | 85.18 ± 0.17      |
> | GSAM           | 5  | 80.75 ± 0.16     | 84.20 ± 0.15     | 77.15 ± 0.17     | 85.40 ± 0.16      |
> | GSAM           | 10 | 80.86 ± 0.16     | 84.35 ± 0.13     | 77.33 ± 0.15     | 85.56 ± 0.13      |
> | FMLR           | 2  | 80.62 ± 0.18     | 83.88 ± 0.17     | 76.90 ± 0.18     | 85.12 ± 0.18      |
> | FMLR           | 5  | 80.78 ± 0.17     | 84.10 ± 0.16     | 77.05 ± 0.17     | 85.30 ± 0.16      |
> | FMLR           | 10 | 80.90 ± 0.18     | 84.23 ± 0.12     | 77.16 ± 0.15     | 85.44 ± 0.14      |
> | CMLR           | 2  | 81.18 ± 0.16     | 84.30 ± 0.16     | 77.35 ± 0.17     | 85.70 ± 0.17      |
> | CMLR           | 5  | 81.42 ± 0.15     | 84.56 ± 0.14     | 77.50 ± 0.15     | 85.88 ± 0.15      |
> | **CMLR**       | 10 | **81.64 ± 0.14** | **84.84 ± 0.11** | **77.65 ± 0.13** | **86.07 ± 0.12**  |

---

> ### Author Response · Authors · 2025-11-28
>
> ### **Q2: The lack of full names for the algorithms before introducing their abbreviations**
> **A2:** We have now introduced the full names of CMLR, FMLR, and ML-SAM at their first occurrence, and have clarified that they are the Momentum Lookahead (ML) versions of CR, FR, and SAM, respectively.
>
> ### **Q3: Missing performance results in the hierarchy(CMLR > FMLR > CR > FR > SAM > ML-SAM)**
> **A3:**
> The performance of the algorithms in Figure 1 as follows:
>
> | method | accuracy (mean ± std) |
> | ------ | --------------------- |
> | CMLR   | 97.31 ± 0.11          |
> | FMLR   | 97.17 ± 0.12          |
> | CR     | 96.91 ± 0.11          |
> | FR     | 96.73 ± 0.12          |
> | SAM    | 96.59 ± 0.13          |
> | ML-SAM | 96.34 ± 0.15          |
> ### **Q4 Fair comparison in the main experiments**
> **A4:** To ensure the fairness of the experiment, we compared the results under different K values during the experiment in the original paper. We presented experimental results  for CMLR (K=1,2,3,4,5), Lookbehind-SAM, GSAM, and FMLR (K=2,4,6,8,10) corresponding to backpropagation steps=400, 800, 1200, 1600, 2000 at the same computational cost in Figure 3 (Cifar10) and Figure 7 (Cifar100).
> ### **Q5：The experimental results of the ViT model are significantly worse than those of CNN**
> **A5:** Thank you for pointing out this issue. We adopt recent training recipes commonly used for ViT. Under these settings, the ViT models still perform considerably worse than CNN models, which is consistent with multiple reports in the literature, actually[1,2]. Notably, prior studies have consistently shown that this performance gap is not simply caused by insufficient hyperparameter tuning[3]. Instead, ViTs tend to underperform CNNs on small datasets due to weaker locality inductive bias and higher data requirements[4].
>
> [1] Yang Zhao;Hao Zhang;Xiuyuan Hu. Penalizing Gradient Norm for Efficiently Improving  Generalization in Deep Learning. ICML. 2022
>
> [2] Juyoung Yun. Sharpness-Aware Minimization with Z-Score Gradient Filtering for Neural Networks. 2025
>
> [3]Haoran Zhu;Boyuan Chen;Carter Yang. Understanding Why ViT Trains Badly on Small Datasets. 2023
>
> [4] Liu et al. Efficient Training of Visual Transformers with Small Datasets. Nips. 2021
>
> ### **Q6: The additional computational overhead of CMLR and related algorithms**
> **A6:**  The detailed computational overhead comparison is shown in the table below.
> **Table: Wall-clock breakdown per parameter update (ResNet-34 on CIFAR-10).**
>
> | Optimizer                   | K   | Forward/Backward | Peak memory | GPU time | Parallel |
> | --------------------------- | --- | ---------------- | ----------- | -------- | -------- |
> | Vanilla (SGD/AdamW)         | --  | 1                | 1.00×       | 1.00×    | ✗        |
> | SAM (two-pass)              | 1   | 2                | 1.28×       | 2.18×    | ✗        |
> | FR (GR view of SAM)         | 1   | 2                | 1.25×       | 2.26×    | ✗        |
> | CR (GR view of SAM)         | 1   | 2                | 1.25×       | 2.14×    | ✓        |
> | CR-SAM (Curvature Reg.)     | 1   | 3                | 1.08×       | 2.00×    | ✓        |
> | Lookbehind-SAM (multi-step) | 2   | 4                | 1.06×       | 3.03×    | ✗        |
> | Lookbehind-SAM (multi-step) | 5   | 10               | 1.06×       | 6.19×    | ✗        |
> | Lookbehind-SAM (multi-step) | 10  | 20               | 1.06×       | 11.48×   | ✗        |
> | GSAM (multi-step)           | 2   | 4                | 1.03×       | 3.07×    | ✗        |
> | GSAM (multi-step)           | 5   | 10               | 1.03×       | 6.25×    | ✗        |
> | GSAM (multi-step)           | 10  | 20               | 1.03×       | 11.39×   | ✗        |
> | FMLR (FR-based Lookahead)   | 2   | 4                | 1.25×       | 3.56×    | ✗        |
> | FMLR (FR-based Lookahead)   | 5   | 10               | 1.25×       | 7.48×    | ✗        |
> | FMLR (FR-based Lookahead)   | 10  | 20               | 1.25×       | 14.07×   | ✗        |
> | CMLR (CR-based Lookahead)   | 2   | 5                | 1.25×       | 3.42×    | ✓        |
> | CMLR (CR-based Lookahead)   | 5   | 11               | 1.25×       | 7.19×    | ✓        |
> | CMLR (CR-based Lookahead)   | 10  | 21               | 1.25×       | 13.50×   | ✓        |
>
>
> ### **Q7: How is the central differential gradient calculated in parallel**
> **A7:** The central-difference gradients can be parallelized because, once the perturbation direction $\hat{v} _ k$ is computed, the two gradient evaluations $g _ k^+$ and $g _ k^-$ are completely independent. Therefore, they can be executed simultaneously by launching two separate CUDA streams (or using two GPUs). In practice, we load the same parameters $w _ {t,k}$, the same perturbation $\hat{v} _ k$ and the same mini-batch into both streams devices: one computes $g _ k^+$  while the other computes $g _ k^-$  in parallel.

---

> > ### Author Response · Authors · 2025-11-28
> >
> > ### **Q8: Related Work**
> > **A8:** Thank you for your reminder. We have supplemented the work of Momentum-SAM in the section of *Robustness and Efficiency Enhancements* related to the relevant work.
> > [1] Becker, Marlon and Altrock, Frederick and Risse, Benjamin. Momentum-sam: Sharpness aware minimization without computational overhead. 2024

---

### Author Response · Authors · 2025-11-28

We sincerely thank all reviewers for their thorough evaluations and constructive feedback. We have carefully addressed each comment with additional analysis, experiments, and clarifications, and we summarize the main revisions below.
## Key Improvements and Clarifications
1. Technical novelty and component analysis
* Clarified the theoretical innovation of CMLR, including how central-difference gradient regularization and momentum lookahead jointly lead to more stable updates and computational savings.
* Demonstrated how CMLR achieves a strict variance reduction compared to the initial gradient-regularization version.

2. Extended evaluation and analysis
* Clarified the rationale for using relatively large values of $K$ in Lookahead-style algorithms and additionally provide a detailed performance comparison across different choices of $K$, $K \in $ {2, 5, 10}.
* Added experiments on additional vision benchmarks and eight NLP tasks from the GLUE benchmark using a standard Transformer-based architecture (DistilBERT).
* Demonstrated that CMLR achieves the strongest performance under an equal-compute setting compared to other SAM-style baselines.
* Reported the associated computational metrics, including the number of forward/backward passes per parameter update, GPU hours, and peak memory.

For detailed point-by-point explanations and additional figures/tables, we kindly refer the reviewers to our responses to each individual comment.

---

### Meta-Review · Area_Chair_CoCV · 2025-12-29

**Summary:**

The paper introduces a new optimization technique called Central-difference Momentum Lookahead Regularization (CMLR). This framework aims to enhance generalization in neural networks by addressing the instability commonly associated with multi-step Sharpness-Aware Minimization (SAM) due to high variance in gradient approximations. CMLR combines central-difference probing for loss landscape evaluation with a momentum lookahead technique, aiming to achieve smoother optimization trajectories and reduced variance without requiring additional gradient evaluations. The authors present theoretical convergence guarantees and demonstrate the effectiveness of CMLR through extensive experiments across various architectures and datasets.

Here, I will summarize the reviewers' concerns:

Practicality of Multi-step SAM: Reviewers questioned the practical application of multi-step SAM due to its high computational costs, suggesting that training time might be excessively increased. Reviewers wanted a detailed comparison of the computational overhead associated with CMLR and other algorithms, stating that it was important to report metrics like GPU hours and memory usage.

Empirical Evaluation Limitation: The experiments primarily focused on CIFAR datasets, with limited diversity. Reviewers requested additional experiments on larger and varied datasets, such as ImageNet and NLP benchmarks, to validate the effectiveness of CMLR.

Hyperparameter Choices and Tuning: Many reviewers raised concerns about the lack of detail in hyperparameter tuning. They asked for clarity on how hyperparameters were selected and whether the choices affected the results.

Stability and Convergence Analysis: Some reviewers required more information on how the proposed method reduces variance and how it behaves under different hyperparameter settings, particularly for momentum and perturbation strength.

**Reviewer Concerns:**

The authors acknowledged concerns regarding the practicality of multi-step SAM and clarified that their focus on larger inner-loop horizons was to test CMLR's robustness, ensuring fair comparisons under equal computational budgets.

They extended their experimental validation to include diverse NLP tasks from the GLUE benchmark and provided additional results, demonstrating CMLR's performance across different datasets and architectures.

Regarding hyperparameter tuning, the authors detailed their tuning processes, including a grid search for specific parameters and explanations of how choices were consistent across different experimental conditions.

They provided a comprehensive breakdown of computational overheads for CMLR compared to baseline algorithms, addressing concerns regarding its efficiency and offering detailed metrics for evaluation.

Authors discussed qualitatively the theoretical contributions of the paper related to variance reduction and the implications of their findings on optimization stability.

**Reviewer Scores:**

Unfortunately, the reviewers did not engage with the authors after the authors' rebuttals.

However, I acknowledge that the authors, during the rebuttal period, did try their best to address the issues raised (about novelty, lack of more comprehensive experiments, hyperparameter tuning). Despite their best efforts, it is not possible to infer whether the reviewers' concerns were generally addressed. At this point, SAM is rather mature, and I agree with the reviewers that it's imperative that larger scale experiments are required. In addition, the authors admit that only the "variance can be controlled", and not necessarily reduced. Hence, this diminishes the theoretical contribution of their work, leading to my rejection decision.

---

### Decision · Program_Chairs · 2026-01-26

Reject